# Mutations and Differential Transcription of Mating-Type and Pheromone Receptor Genes in *Hirsutella sinensis* and the Natural *Cordyceps sinensis* Insect-Fungi Complex

**DOI:** 10.3390/biology13080632

**Published:** 2024-08-18

**Authors:** Xiu-Zhang Li, Meng-Jun Xiao, Yu-Ling Li, Ling Gao, Jia-Shi Zhu

**Affiliations:** State Key Laboratory of Plateau Ecology and Agriculture, Qinghai Academy of Animal Science and Veterinary, Qinghai University, Xining 810016, China; xiuzhang11@163.com (X.-Z.L.); 15574237597@163.com (M.-J.X.); yulingli2000@163.com (Y.-L.L.); silvering2008@163.com (L.G.)

**Keywords:** differential occurrence and transcription of mating-type genes, alternative splicing of the MAT1-2-1 gene, genetic and transcriptional reproduction regulation, mutant MAT1-1-1 protein, mutant α-pheromone receptor protein, genotypes of *Ophiocordyceps sinensis*, homothallism, pseudohomothallism, physiological heterothallism, hybridization

## Abstract

**Simple Summary:**

The natural *Cordyceps sinensis* insect–fungal complex is one of the most highly valued therapeutic agents in traditional Chinese medicine. The sexual reproduction of *Ophiocordyceps sinensis* is controlled by the expression of mating-type genes within the *MAT1-1* and *MAT1-2* idiomorphs and a-/α-pheromone receptor genes. This study revealed the differential expression of these genes in *Hirsutella sinensis* (Genotype #1 among the 17 genotypes of *O. sinensis*) and alternative splicing of the MAT1-2-1 gene, characterized by an unspliced intron I that contains stop codons. Mutations of the mating and pheromone receptor proteins were also identified, potentially leading to changes in the secondary structures and functionalities of the proteins. These results indicate the requirement of mating partners under either heterothallism or hybridization during the sexual life of natural *C. sinensis* but do not support the “self-fertilization” hypothesis for *O. sinensis* under homothallism or pseudohomothallism.

**Abstract:**

Sexual reproduction in ascomycetes is controlled by the mating-type (MAT) locus. (Pseudo)homothallic reproduction has been hypothesized on the basis of genetic data from *Hirsutella sinensis* (Genotype #1 of *Ophiocordyceps sinensis*). However, the differential occurrence and differential transcription of mating-type genes in the *MAT1-1* and *MAT1-2* idiomorphs were found in the genome and transcriptome assemblies of *H. sinensis*, and the introns of the MAT1-2-1 transcript were alternatively spliced with an unspliced intron I that contains stop codons. These findings reveal that *O. sinensis* reproduction is controlled at the genetic, transcriptional, and coupled transcriptional-translational levels. This study revealed that mutant mating proteins could potentially have various secondary structures. Differential occurrence and transcription of the a-/α-pheromone receptor genes were also found in *H. sinensis*. The data were inconsistent with self-fertilization under (pseudo)homothallism but suggest the self-sterility of *H. sinensis* and the requirement of mating partners to achieve *O. sinensis* sexual outcrossing under heterothallism or hybridization. Although consistent occurrence and transcription of the mating-type genes of both the *MAT1-1* and *MAT1-2* idiomorphs have been reported in natural and cultivated *Cordyceps sinensis* insect-fungi complexes, the mutant MAT1-1-1 and α-pheromone receptor transcripts in natural *C. sinensis* result in N-terminal or middle-truncated proteins with significantly altered overall hydrophobicity and secondary structures of the proteins, suggesting heterogeneous fungal source(s) of the proteins and hybridization reproduction because of the co-occurrence of multiple genomically independent genotypes of *O. sinensis* and >90 fungal species in natural *C. sinensis*.

## 1. Introduction

Natural *Cordyceps sinensis* is one of the most highly valued therapeutic agents in traditional Chinese medicine and has a rich history of clinical use in health maintenance, disease amelioration, postdisease and postsurgery recovery, and anti-aging therapy [1,2,3]. The Chinese Pharmacopoeia defines natural *C. sinensis* as an insect-fungal complex containing the *Ophiocordyceps sinensis* fruiting body and the remains of a Hepialidae moth larva (an intact, thick larval body wall with numerous bristles, an intact larval intestine and head tissues, and fragments of other larval tissues) [4,5,6]. Histological, mycological, and molecular biological studies of natural *C. sinensis* have revealed 17 genomically independent genotypes of *O. sinensis* [5,6,7,8,9,10,11,12,13,14,15,16] and more than 90 species spanning at least 37 fungal genera and larval genes, demonstrating its multicellular heterokaryotic structure and genetic heterogeneity [5,6,9,14,15,16,17,18,19,20,21,22,23,24,25,26]. The expression of mating-type genes during the sexual life of natural *C. sinensis* is much more complex than that in pure fungal cultures and is mutually and/or antagonistically disrupted by the expression of the metagenomes of multiple co-colonized fungi and by larval host innate immunity and acquired immunological responses during *O. sinensis* fungal infection and proliferation [5,6,18,26,27,28]. It has been suggested that larval tissues are not just culture media that passively provide nutrients for fungal growth, similar to the media used in in vitro culture or fermentation, but also play active defensive roles against fungal infection and growth [5,6,28,29]. Notably, the Latin name *Cordyceps sinensis* has been used indiscriminately since the 1840s for both the teleomorph/holomorph of *C. sinensis* fungus and the wild insect–fungal complex, and the fungus was renamed *O. sinensis* in 2007 [4,5,6,30,31]. In this paper, we refer to the fungus *Hirsutella sinensis* (Genotype #1) and genetically related Genotypes #2–17 fungi as *O. sinensis* (note that the 17 genomically independent genotypes share a common genetic ancestor [8], but the taxonomic positions of Genotypes #2–17 have not been determined), and we continue the customary use of the name *C. sinensis* to refer to the wild or cultivated insect-fungi complex, although this practice will likely be replaced by the discriminative use of exclusive Latin names in the future.

The sexual reproductive behavior of ascomycetes is controlled by transcription factors encoded at the mating-type (*MAT*) locus [32,33,34,35]. Hu et al. [36] detected the MAT1-1-1 and MAT1-2-1 genes of the *MAT1-1* and *MAT1-2* idiomorphs in the genome assembly ANOV00000000 of the *H. sinensis* strain Co18 and proposed a self-fertilization hypothesis for *H. sinensis* (GC-biased Genotype #1 of *O. sinensis*) under homothallism. The MAT1-1-1 gene encodes a protein possessing a conserved α-box domain, and the MAT1-2-1 gene encodes a protein possessing a high-mobility-group (HMG) box domain [34,35]. Bushley et al. [37] described the multicellular heterokaryotic structures of natural *C. sinensis* hyphae and ascospores containing mononucleated, binucleated, trinucleated, and tetranucleated cells; in addition, they detected the MAT1-1-1, MAT1-1-2, and MAT1-1-3 genes of the *MAT1-1* idiomorph and the MAT1-2-1 gene of the *MAT1-2* idiomorph of *H. sinensis*, which are located >4 kb apart in the *H. sinensis* strain 1229 genome. On the basis of these results, they hypothesized that *H. sinensis* underwent pseudohomothallism. However, Zhang et al. [38] and Zhang and Zhang [39] reported differential occurrence of the MAT1-1-1 and MAT1-2-1 genes in various *C. sinensis* isolates and hypothesized facultative hybridization for *O. sinensis*. These hypotheses regarding the sexual reproductive behavior of *O. sinensis* were proposed based on the genetic information of *H. sinensis* without considering the entire expression process of the *H. sinensis* mating-type genes, including genetic, epigenetic, transcriptional, posttranscriptional, translational, and posttranslational modifications, as well as the activation and degradation of mating-type proteins.

Many studies on natural *C. sinensis* and *O. sinensis* fungi have focused primarily on *H. sinensis*, which has been postulated to be the only anamorph of *O. sinensis* [40]. Following this sole anamorph hypothesis, the *H. sinensis* strain EFCC7287, Genotype #1 of *O. sinensis*, was selected for use as a reference species for the renaming of *Cordyceps sinensis* to *Ophiocordyceps sinensis* [30]. However, this renaming did not involve the natural insect-fungi complex and did not cover the remaining 16 genotypes of *O. sinensis* fungi, for which pure cultures have not been available for taxonomy and nomenclature multigene studies or for genomics, transcriptomics, proteomics, natural chemistry, and pharmacology studies [5,7,8,9,10,11,12,13,14,15,41,42]. Numerous studies have reported that the sequences of *O. sinensis* Genotypes #2–17, regardless of whether they are GC- or AT-biased, are not found in the genome assemblies ANOV00000000, JAAVMX000000000, LKHE00000000, LWBQ00000000, and NGJJ00000000 of *H. sinensis* (GC-biased Genotype #1) strains Co18, IOZ07, 1229, ZJB12195, and CC1406-203, respectively [36,43,44,45,46], providing evidence of the genomic independence of the 17 *O. sinensis* genotypes, which are interindividual *O. sinensis* fungi [5,8,9,10,11,12,14,15,47,48]. In this paper, we continue using the anamorphic name *H. sinensis* for Genotype #1 of *O. sinensis*, although a group of mycologists [49] have improperly implemented the “One Fungus=One Name” nomenclature rule of the International Mycological Association while disregarding the presence of multiple genomically independent genotypes of *O. sinensis* fungi and inappropriately replacing the anamorphic name *H. sinensis* with the teleomorphic name *O. sinensis* [4,5,6].

The lifecycle of *C. sinensis* includes asexual and sexual growth stages during its development and maturation. *O. sinensis* infection of the larvae of the Hepialidae family initiates the lifecycle of natural *C. sinensis*, which includes several developmental phases (Appendix A [50]): (1) the formation of a stromal primordium; (2) immature *C. sinensis* with a short stroma (1–2 cm) (*O. sinensis* grows asexually in the first two phases); (3) maturing *C. sinensis* in the transition from asexual to sexual growth with an approximately 3–5 cm stroma without the formation of an expanded fertile portion close to the tip of the stroma; and (4) mature *C. sinensis* that grows sexually with a long stroma (usually >5 cm) with an expanded fertile portion close to the stromal tip, densely covering numerous ascocarps before and after ascospore ejection.

*C. sinensis* maturation is associated with dramatic changes in the mycobiota, metagenome, metatranscriptomic and proteomic expression, chemical constituent fingerprint, and pharmacological functions [5,6,11,15,22,45,50,51]. Li et al. [14,15] reported that naturally and semiejected *C. sinensis* ascospores are associated with the differential co-occurrence of several GC- and AT-biased genotypes of *O. sinensis*. The biomasses/abundances of the GC- and AT-biased genotypes of *O. sinensis* change dynamically in an asynchronous, disproportional manner in the caterpillar body, stroma, stromal fertile portion (with ascocarps), and ascospores of natural *C. sinensis* during maturation [6,9,11,14,15]. Thus, identifying the maturation stages of natural *C. sinensis* specimens during the examination of mating-type gene expression is essential.

Numerous genetic, genomic, and transcriptomic sequences of *H. sinensis* strains and natural *C. sinensis* specimens are available in the GenBank database, which enables further bioinformatic examination of the hypotheses regarding the sexual reproduction strategy of *O. sinensis* at the genome, transcriptome, and protein levels. In addition to the genetic occurrence of mating-type genes in 237 *H. sinensis* strains, this paper discusses the differential transcription and alternative splicing of mating-type genes and pheromone receptor genes, as well as the mutations and related conformational alterations of their encoded proteins involved in *O. sinensis* sexual reproduction in *H. sinensis* and natural *C. sinensis*.

## 2. Materials and Methods

### 2.1. Gene and Genome Sequences of H. sinensis Strains and Natural C. sinensis

Appendix A lists the 237 *H. sinensis* strains that were used to obtain the mating-type gene sequences, which are available in GenBank and the original papers [10,36,37,38,39,43,44,45,46,48,52]. Five sets of genome assemblies—ANOV00000000, JAAVMX000000000, LKHE00000000, LWBQ00000000, and NGJJ00000000—of the *H. sinensis* strains Co18, IOZ07, 1229, ZJB12195, and CC1406-20395, respectively, are available in GenBank [36,43,44,45,46]. Table 1 lists the sequencing and assembly technologies (software) for the 5 genome assemblies.

### 2.2. Transcriptome and Metatranscriptome Assemblies and Transcripts of the Mating-Type Genes of H. sinensis Strains and Natural C. sinensis

Table 1 also lists the sequencing and assembly technologies (software) for the transcriptome assembly (GCQL00000000) of the *H. sinensis* strain L0106 and the metatranscriptome assembly (GAGW00000000) for the natural *C. sinensis* specimen collected from Kangding County, Sichuan Province, China [53,54].

The mycelia of the *H. sinensis* strain L0106 were collected for total RNA extraction from cultures grown for 3, 6, and 9 days [52]. The total RNA (20 mg per sample) was subjected to mRNA purification, and the total mRNA was used to construct a cDNA library for sequencing.

A natural *C. sinensis* sample (unknown maturational status) was collected from Kangding County, Sichuan Province, China [53]. The total RNA from this sample was sequenced (Table 1).

Another metatranscriptome assembly was derived from specimens of mature natural *C. sinensis* that were collected from Deqin, Yunnan Province, China (*cf*. the Appendix of [54]). The total RNA was extracted from the fruiting bodies to construct a cDNA library (Table 1). The sequencing reads were deposited in GenBank under BioProject #PRJNA382001 and assembled via Trinity (version r20140717) [55]. The metatranscriptome assembly sequences were uploaded to the data repository, www.plantkingdomgdb.com/Ophiocordyceps_sinensis/data/cds/Ophiocordyceps_sinensis_CDS.fas (accessed on 18 January 2018), which is currently inaccessible, but a previously downloaded cDNA file was used for bioinformatic analysis.

### 2.3. Sequence Alignment Analysis

We reanalyzed all gene, genome, transcript, transcriptome, and metatranscriptome sequences and other PCR-amplified DNA sequences of the mating-type and pheromone receptor genes of *H. sinensis* and natural *C. sinensis* via the MegaBlast, discontinuous MegaBlast, Blastp, Tblastn, or blastn suite-SRA programs provided by GenBank (https://blast.ncbi.nlm.nih.gov/) (frequently accessed) to compare and align the nucleotide and amino acid sequences.

### 2.4. Amino Acid Property and Scale Analyses

The overall hydrophobicity/hydrophilicity of the mating-type and pheromone receptor proteins of *H. sinensis* and natural *C. sinensis* were characterized on the basis of the chemical-physical properties of the component amino acids via Peptide 2.0 (https://www.peptide2.com/N_peptide_hydrophobicity_hydrophilicity.php) (accessed on 2 March 2024). The component amino acids were scaled (Appendix A) and plotted sequentially at a window size of 21 amino acid residues for the α-helices, β-sheets, β-turns, and coils of the proteins via the linear weight variation model of the ExPASy ProtScale algorithm (https://web.expasy.org/protscale/) (accessed on 2 March 2024) provided by the SIB Swiss Institute of Bioinformatics [56,57,58]. The overall amino acid hydrophobicity properties and the waveforms of the ProtScale plots for the proteins were compared to explore potential alterations in the secondary structures of the mating-type and pheromone receptor proteins.

## 3. Results

### 3.1. Mating-Type Genes and Encoded Proteins in H. sinensis Strains

Intraspecific variations in 237 *H. sinensis* strains and wild-type *C. sinensis* isolates that reportedly differentially contain the MAT1-1-1 and/or MAT1-2-1 genes were found in GenBank (Appendix A) [36,37,38,39,43,44,45,46,48,49].

Three mating-type genes of the *MAT1-1* idiomorph have been previously identified in the genome sequence KC437356 of the *H. sinensis* strain 1229 by Bushley et al. [37]: the MAT1-1-1 gene (6530→7748), MAT1-1-2 gene (4683→6183), and MAT1-1-3 gene (3730→4432). The sequences of these mating-type genes from KC437356 show 99.9–100% homology with the segment sequences of the genome assemblies LKHE01001116 (3691←4909; 5374←6874; 7125←7827) of strain 1229; ANOV01017390 (302←1519) and ANOV01017391 (276←1776; 2027←2729) of strain Co18; and JAAVMX010000001 (6,698,911→6,700,129; 6,696,939→6,698,439; 6,695,986→6,696,688) of strain IOZ07 (Table 2) [36,37,38,39,43,46]. Note that the arrows “→” and “←” indicate sequences in the sense and antisense chains of the genomes, respectively. However, the mating-type genes of the *MAT1-1* idiomorph are absent from the genome assemblies NGJJ00000000 and LWBQ00000000 of the *H. sinensis* strains CC1406-203 and ZJB12195, respectively [44,45]. The MAT1-1-1 gene was detected in 127 *H. sinensis* strains, with 98.0–100% homology among the sequences, but no MAT1-1-2 or MAT1-1-3 genes were detected in these strains [5,10,36,37,38,39,48,49].

The MAT1-2-1 sequence JQ325153 of the *H. sinensis* strain GS09_121 [59] is highly homologous with the genome segment sequences ANOV01000063 (9319→10,191), LKHE01001605 (13,851←14,723), LWBQ01000021 (238,864←239,7), and NGJJ01000619 (23,021←23,89336) of the *H. sinensis* strains Co18, 1229, ZJB12195, and CC1406-203, respectively (Table 2) [36,43,44,45]. The highly homologous MAT1-2-1 gene was identified in 183 other *H. sinensis* strains [10,36,37,38,39,48,49,59]. However, the MAT1-2-1 gene was absent in the genome assembly JAAVMX000000000 of strain IOZ07 [46].

The presence of multicellular heterokaryotic structures in *C. sinensis* hyphae and ascospores was detected via fluorescence microscopy techniques in natural *C. sinensis* by Bushley et al. [37]. These authors also reported that the *MAT1-1* and *MAT1-2* idiomorphs are not closely linked because the MAT1-1-1 and MAT1-2-1 genes are located more than 4 kb apart in the *H. sinensis* genome; the authors proposed that *H. sinensis* undergoes pseudohomothallism. We found that the MAT1-1-1 and MAT1-2-1 genes are located on contig_17390 of ANOV01017390 (302←1519) and contig_63 of ANOV01000063 (9319→10,191) in the genome of strain Co18 [36] and on scaffold_1116 of LKHE01001116 (3691←4909) and scaffold_1605 of LKHE01001605 (13,851←14,723) in the genome of strain 1229 [43]. Table 2 summarizes the co-occurrence and differential occurrence of the MAT1-1-1 and MAT1-2-1 genes in the genome assemblies of *H. sinensis* strains 1229, CC1406-203, Co18, IOZ07, and ZJB12195 [36, 43–46].

Notably, Zhang and Zhang [39] reported 4.9% and 6.1% allelic variation (93.9% and 95.1% similarity) in the full-length sequences of the MAT1-1-1 and MAT1-2-1 genes, respectively, and 4.7% and 5.7% variation in the exon sequences of numerous *C. sinensis* isolates, which might disrupt the translation of the coding sequences of the genes.

### 3.2. Differential Transcription and Alternative Splicing of Mating-Type Genes in the Transcriptome of H. sinensis Strains

In addition to the occurrence of mating-type genes in the genome of *H. sinensis*, transcription of the mating-type genes of both the *MAT1-1* and *MAT1-2* idiomorphs and the production of fully activated encoded proteins within a single cell are needed to accomplish the homothallic mating process and to control the development and maturation of fruiting bodies, ascocarps, and ascospores [32,33,34,35,60]. Table 2 shows the absence of the sequences of the MAT1-1-1, MAT1-1-2, and MAT1-1-3 genes in the transcriptome assembly GCQL00000000 of the *H. sinensis* strain L0106 [52]. Therefore, the differential transcription of the genes of the *MAT1-1* idiomorph may constitute a mechanism of sexual reproduction control in addition to the differential occurrence of the genes in *H. sinensis* strains, as shown in Appendix A.

The MAT1-2-1 gene JQ325153 of the *H. sinensis* strain GS09_121 contains two introns: “Intron Phase-Two” for intron I (52 nucleotides [nt]; 187→238 of JQ325153) and “Intron Phase-One” for intron II (55 nt; 535→589 of JQ325153). Both intron phases disrupt the open reading frame and codons (Appendix A) [37,38,39,48,59]. Appendix A also shows 3 other MAT1-2-1 genes (FJ654187, JQ325237, and KM197536) from the *H. sinensis* strains CS2, XZ12_16, and XZ12_16, respectively, which were 97.7–99.5% similar to the sequence of JQ325153. A single T-to-C transition downstream of the third stop codon was present in intron I in the sequences of JQ325237 and KM197536 and apparently had no impact on the three upstream stop codons in intron I.

The MAT1-2-1 gene JQ325153 contains 3 exons (388←671; 672←967; 968←1153) and exhibits 99.7–100% homology with the transcriptome assembly GCQL01020543 of the *H. sinensis* strain L0106 [39,52]. Xia et al. [54] conducted a metatranscriptome study of natural *C. sinensis* and confirmed that the MAT1-2-1 transcript sequence (OSIN7649) is spliced from both introns I and II in the metatranscriptome assembly (further discussion below). Bushley et al. [37] reported “only one spliced intron of 55 bp” when comparing “DNA and cDNA sequences of MAT1-2-1” in the *H. sinensis* strain 1229, which “did not result from contamination of genomic DNA”, and illustrated the alternatively spliced MAT1-2-1 cDNA with unspliced intron I and spliced intron II. Appendix A shows the MAT1-2-1 sequences, which possess 3 stop codons, “TAA” or “TGA”, within intron I (the first, third, and 13th triplets), leading to translational arrest of the MAT1-2-1 transcript with unspliced intron I.

A BLAST search of the GenBank database revealed that the intron I sequence of the MAT1-2-1 gene is 100% identical to the MAT1-2-1 gene segments of the genome assemblies LKHE01001605, NGJJ01000619, ANOV01000063, and LWBQ01000021 of strains 1229, CC1406-203, Co18, and ZJB12195, respectively, and to the DNA sequences of more than 100 *H. sinensis* strains. These findings exclude the possibility that gene mutations within MAT1-2-1 intron I occur to avoid translational interruption caused by the presence of stop codons within intron I. A five-base insertion occurs in intron I of the MAT1-2-1 genes FJ654204 and FJ654205 in two *H. sinensis* isolates—XZ-LZ07-H1 and XZ-LZ07-H2—obtained from the Nyingchi District in Tibet, which was postulated to be the center of origin of the species *H. sinensis* [10]. The insertion is located downstream of the first stop codon, “TAA”. Thus, the alternatively spliced MAT1-2-1 gene with unspliced intron I might represent an additional mechanism of the coupled transcriptional-translational regulation of the mating process of *H. sinensis* strain 1229 [37], which is distinct from the gene silencing used for transcriptional control (Table 2), although the full expression process (genetic, epigenetic, transcriptional, posttranscriptional, translational, and posttranslational modifications; protein activation-degradation; etc.) of the *H. sinensis* MAT1-2-1 gene needs to be thoroughly examined in future reproductive physiology studies.

The above transcriptome analysis of *H. sinensis* strains 1229 and L0106 [37,52] revealed the transcriptional and coupled transcriptional-translational controls of mating-type gene expression in *H. sinensis*. These results are inconsistent with the hypotheses of self-fertilization under (pseudo)homothallism for *H. sinensis* [36,37]. Instead, these findings suggest that *O. sinensis* requires mating partners from *H. sinensis*, monoecious or dioecious, for physiological heterothallic reproduction or from heterospecific fungal species to ensure hybridization if the species are able to overcome interspecific reproductive isolation.

The available evidence regarding mating-type gene transcription might explain why previous efforts to cultivate pure *H. sinensis* in research-oriented academic settings to induce the production of fruiting bodies and ascospores have consistently failed [37,49,61,62]. The evidence may also explain why a successful inoculation–cultivation project in a product-oriented industrial setting presented a species contradiction between the GC-biased Genotype #1 *H. sinensis* strains used as anamorphic inoculants and the teleomorphic AT-biased Genotype #4 of *O. sinensis* in the fruiting body of cultivated *C. sinensis*, as reported by Wei et al. [63].

### 3.3. Differential Occurrence of Mating-Type Genes in Natural and Cultivated C. sinensis Insect-Fungi Complexes

The above sections discuss the differential occurrence and transcription of mating-type genes in pure *H. sinensis* strains and wild-type *C. sinensis* isolates. Mycobiota studies have demonstrated the differential coexistence of >90 co-colonized fungi belonging to at least 37 fungal genera in the stromata and caterpillar bodies of natural *C. sinensis* [23,25], which has been confirmed by metagenomic and metatranscriptomic studies and by the identification of >100 mitogenomic transcript repeats in natural and cultivated *C. sinensis* [18,27,29,53,54,64,65,66,67]. Many studies have reported differential co-occurrences of several GC- and AT-biased genotypes of *O. sinensis* in different compartments (the stroma, caterpillar body, and stromal fertile portion densely covered with numerous ascocarps and ascospores) of natural *C. sinensis*, the sequences of which reside in the independent fungal genomes of different *O. sinensis* fungi [5,6,7,8,9,10,11,12,13,14,15,16,41,42,47,63]. Notably, the indiscriminate use of the Latin name *Ophiocordyceps sinensis* for multiple genotypes of *O. sinensis* fungi and natural and cultivated insect-fungi complexes has prompted scientists to cautiously distinguish the study materials to ensure the correct wording and understanding of research papers/reports [5,6,13,14,47].

The mating-type genes of both the *MAT1-1* and *MAT1-2* idiomorphs reportedly exist in natural and cultivated *C. sinensis*, for example, (1) in the metagenome of mature natural *C. sinensis* [54]; (2) in the compartments of natural *C. sinensis* at different developmental stages [65]; (3) in cultivated *C. sinensis* at different developmental phases [18]; and (4) in the early-developed stroma and caterpillar body of natural *C. sinensis* with very low read count values and in 31 other *C. sinensis* specimens [66]. However, the detection of the mating-type genes of both the *MAT1-1* and *MAT1-2* idiomorphs reported in these studies might not correctly indicate the capability at the genetic level for self-fertilization of a single *H. sinensis* cell because of the differential coexistence of multiple co-colonized fungi and multiple genotypes of *O. sinensis* in the compartments of natural and cultivated *C. sinensis* [5,6,7,8,9,10,11,12,13,14,15,16,18,39,47,63,64,65,66,67]. Hence, the possibility of physiological heterothallic outcrossing or hybridization should not be neglected, particularly because of the species contradiction between the anamorphic inoculants of GC-biased *H. sinensis* strains used in industrial cultivation projects and the sole teleomorph of AT-biased Genotype #4 in cultivated *C. sinensis* [63].

### 3.4. Differential Transcription of Mating-Type Genes in Natural and Cultivated C. sinensis Insect-Fungi Complexes

#### 3.4.1. Transcription of Mating-Type Genes in the Metatranscriptome Assemblies of Natural *C. sinensis*

Xiang et al. [53] reported a metatranscriptome project with a natural *C. sinensis* sample (unknown maturation stage) collected from Kangding County, Sichuan Province, China. The assembled metatranscriptome segment 297←1129 of GAGW01008880 aligns with the MAT1-1-1 gene sequence KC437356 but is shorter because of the absence of exon I, intron I, and 48 bp of exon II of the MAT1-1-1 gene (Appendix A). Its intron II (48 bp) is spliced between transcript nucleotides 358 and 359. Excluding the missing segments, GAGW01008880 is 100% identical to the remaining coding sequences (771 bp of exon II and the full length of exon III) of the MAT1-1-1 gene.

Xia et al. [54] reported a metagenome-metatranscriptome project with fully mature *C. sinensis* samples collected from Deqin County, Yunnan Province, China. The metatranscriptome assembly OSIN7648 in the Appendix of [54] shares 87.3% similarity with the MAT1-1-1 gene sequence KC437356 and possesses three spliced/deleted segments, introns I and II (52 bp and 48 bp in pink in Appendix A, respectively), and an additional deleted segment of 54 bp (in blue in Appendix A) within exon II. Excluding the three spliced/deleted segments, OSIN7648 shares 100% identity with the coding sequences of exons I and III and the remaining exon II.

Compared with the coding sequence (1119 bp) of the MAT1-1-1 gene of KC437356, the MAT1-1-1 transcripts in both metatranscriptome assemblies of natural *C. sinensis* are short [37,53,54]. OSIN7648 (1065 bp) contains an additional deleted segment (54 bp in blue in Appendix A) within exon II of the MAT1-1-1 gene [54], which may produce a middle-truncated MAT1-1-1 protein that is missing 18 aa residues. Compared with the MAT1-1-1 gene, GAGW01008880 is 286 bp shorter because of the absence of a 238-bp segment of exon I and a 48-bp segment of exon II [53] and may produce an N-terminal-truncated MAT1-1-1 protein. Appendix A also shows a 296-bp noncoding segment of GAGW01008880 flanking downstream of the stop codon “TAG”.

The transcript sequences of the MAT1-1-2 and MAT1-1-3 genes were not identified in the metatranscriptome assemblies of the natural *C. sinensis* samples [53,54].

The MAT1-2-1 gene sequence was not detected in the metatranscriptome assembly GAGW00000000 of natural *C. sinensis* (unknown maturation stage) collected from Kangding County, Sichuan Province (*cf*. Appendix A) [53]. However, the MAT1-2-1 gene sequence was found in the metatranscriptome assembly OSIN7649 of mature natural *C. sinensis* collected from Deqin County, Yunnan Province, which shares 100% homology with the transcriptome assembly GCQL01020543 of the *H. sinensis* strain L0106 [52,54].

#### 3.4.2. Transcription of Mating-Type Genes in Unassembled Metatranscriptome Sequencing Read Archives of Natural and Cultivated *C. sinensis*

Zhang et al. [64] conducted a metatranscriptome study in natural *C. sinensis* at the early maturing stage and reported differential transcription of mating-type genes. The MAT1-1-1 gene (OCS_06642) was reportedly transcribed at a similar level in all five samples collected from the counties of Qumalai (specimen #1), Gonghe (specimen #2), and Zaduo (specimen #3) of Qinghai Province and from Naqu (specimen #4) and Nyingchi (specimen #5) in Tibet. However, the heatmap shown in Figure 3B of [64] demonstrated very different transcription in all five specimens. The MAT1-1-3 gene (OCS_06644) was transcribed at a low level only in specimen #4 but was not transcribed in the other specimens, whereas the MAT1-2-1 gene (OCS_00196) was transcribed at low levels in specimens #4 and #5 but was not transcribed in specimens #1–3.

Zhong et al. [65] conducted another metatranscriptome study of the early-stage stroma and caterpillar body of natural *C. sinensis* collected from Yushu, Qinghai Province, China, and observed differential transcription of mating-type genes. They reported that “the MAT1-1-1 gene could be detected in all three groups of samples” but did not report whether the transcript or cDNA of the MAT1-1-1 gene was detected in parallel. The authors also reported that “MAT1-1-2 was not detected in all samples”. They identified the transcripts of the MAT1-1-3 and MAT1-2-1 genes at “a very low read count value” but were unsure whether they “might participate in fruiting body development” in natural *C. sinensis*. Thus, further studies aiming to validate the transcription of mating-type genes and the functions of mating proteins in natural *C. sinensis* at different developmental stages are needed.

Li et al. [18] conducted a metatranscriptome study in cultivated *C. sinensis* at all developmental phases and observed the differential transcription of the mating-type genes of the *MAT1-1* and *MAT1-2* idiomorphs at very low levels (with read counts of 0–40 and fragments per kilobase of exon model per million mapped fragments [FPKM] values of 0–8.4) (note: an FPKM value less than 10 indicates very low levels of transcription). The mating-type genes of the *MAT1-1* idiomorph were not transcribed in the hyphae (read count = 0) but were differentially transcribed at very low levels in cultivated *C. sinensis* at other developmental stages. The MAT1-2-1 gene was transcribed at very low levels (FPKM 0.16–0.93) in cultivated *C. sinensis* at all developmental stages.

Zhao et al. [66] conducted a metatranscriptome study in cultivated *C. sinensis* and reported nearly no transcription of the mating-type genes of the *MAT1-1* and *MAT1-2* idiomorphs (transcripts per million reads [TPM] of 0–2.27). The authors presumed that these genes might not play a role in the initiation of fruiting body development in cultivated *C. sinensis*.

We conducted a BLAST search of the GenBank database against the unassembled transcript reads as part of (1) the metatranscriptome BioProject PRJNA325365 and the Sequence Read Archive (SRA) SRP076425 of natural *C. sinensis* [64]; (2) the SRA SRP103894 of natural *C. sinensis* at an early developmental stage [65]; (3) the BioProject GSE123085 of cultivated *C. sinensis* at all developmental phases [18]; and (4) the BioProject PRJNA600609 of cultivated *C. sinensis* at the initiation stage [66]. We identified hundreds to thousands of unassembled transcript reads of MAT1-1-1, MAT1-1-2, and MAT1-2-1 cDNA. These unassembled transcript reads share 73.8–100% similarity with the coding sequences of the *H. sinensis* MAT1-1-1, MAT1-1-2, and MAT1-2-1 genes. No match was found between sequences of the *H. sinensis* MAT1-1-3 gene and the unassembled transcript reads of natural and cultivated *C. sinensis*. This range of variation between the sequences of the *H. sinensis* mating-type genes and the unassembled transcript reads indicates multiple fungal sources of the transcripts in natural and cultivated *C. sinensis*.

This section of the bioinformatic study discusses the coexistence of multiple mating-type genes at the metatranscriptome level in natural and cultivated *C. sinensis* [18,53,54,64,65,66]. The differential transcription of the mating-type genes of both the *MAT1-1* and *MAT1-2* idiomorphs in natural and cultivated *C. sinensis*, which include >90 fungal species spanning at least 37 genera and multiple GC- and AT-biased genotypes of *O. sinensis*, provides insufficient evidence that *H. sinensis* is the source of the transcripts and therefore does not sufficiently support the hypothesis that *H. sinensis* uses homothallic/pseudohomothallic mating strategies [5,15,18,36,37,38,39,49,53,54,58,64,65,66].

### 3.5. Variations in Mating Proteins

The above sections explore the differential occurrence and transcription of the mating-type genes of the *MAT1-1* and *MAT1-2* idiomorphs at the genome/metagenome and transcriptome/metatranscriptome levels and discuss the genetic, transcriptional, and coupled transcriptional-translational controls of the mating processes of *H. sinensis* in natural and cultivated *C. sinensis*. Allelic mutations in mating-type genes and their transcripts might affect protein translation and posttranslational processes.

The MAT1-1-1 protein sequences derived from the gene sequences of 127 *H. sinensis* strains show 98.1–100% nucleotide homology [10,35,36,37,38,39,49,59]. Among these, 29 sequences contained one to seven conserved and/or nonconserved mutated residues of a total of 372 aa residues at single-variant sites (Figure 1), sharing 98.1–99.7% similarity with the MAT1-1-1 protein sequence AGW27560 of strain CS68-2-1229. The MAT1-1-1 protein sequence (354 aa) translated from the metatranscriptome assembly OSIN7648 obtained from natural *C. sinensis* specimens collected from Deqin County, Yunnan Province, China, shows 92.5–94.9% similarity to the MAT1-1-1 protein sequence (372 aa) of *H. sinensis*, with a substitution of leucine (L)→phenylalanine (F) and a deletion of 18 aa (SMQREYQAPREFYDYSVS) (Figure 1 and Appendix A) [36,37,38,39,54,59]. This mutation likely produces a middle-truncated MAT1-1-1 protein.

The MAT1-1-1 protein (276 aa) translated from the metatranscriptome assembly GAGW01008880 obtained from a natural *C. sinensis* sample collected from Kangding County of Sichuan Province, China, is 96 aa shorter (MTTRNEVMQRLSSVRADVLLNF-LTDDAIFQLASRYHESTTEADVLTPVSTAAASR-ATRQTKEASCDRAKRPLNAFMAFR-SYYLKLFPDVQQQKTASG) at its N-terminus (Figure 1 and Appendix A) [36,37,38,39,53,59]. This mutation likely produces an N-terminal-truncated MAT1-1-1 protein.

The absence of the MAT1-1-1 transcript in the transcriptome of the *H. sinensis* strain L0106 and the remarkable variations in the MAT1-1-1 protein conformation in natural *C. sinensis* as translated from the MAT1-1-1 transcripts in the metatranscriptome assemblies may suggest heterogeneous fungal sources of the protein in natural insect-fungi complexes (Table 2) [52,53,54].

Table 3 lists the four types of amino acids in MAT1-1-1 proteins on the basis of their chemical-physical and hydrophobic properties determined via Peptide 2.0 [68,69]. Compared with the percentage of amino acids in the MAT1-1-1 protein AGW27560 of the *H. sinensis* strain CS68-2-1229, changes in the amino acid content of the MAT1-1-1 protein sequences translated from the metatranscriptome assemblies GAGW01008880 and OSIN7648 of natural *C. sinensis* indicated an altered overall hydrophobicity-hydrophilicity of the proteins [37,53,54].

Figure 2 shows ExPASy ProtScale plots for the α-helices (Panel 2A), β-sheets (Panel 2B), β-turns (Panel 2C), and coils (Panel 2D) of the following MAT1-1-1 proteins: the authentic MAT1-1-1 protein AGW27560 of the *H. sinensis* strain CS68-2-1229 and the middle- and N-terminal-truncated proteins (OSIN7648 and GAGW01008880, respectively) of natural *C. sinensis*. The variable segment region of the MAT1-1-1 protein is outlined with open boxes in green in all plots of Figure 2. Substantial waveform changes are present in all OSIN7648 plots. The overall alterations in hydrophobicity (Table 3) and the α-helix, β-sheet, β-turn, and coil plots (Figure 2) of the MAT1-1-1 proteins indicate variable protein folding and secondary structures, indicating disturbed functionality in the fungal mating process and, in turn, the sexual reproduction of *O. sinensis*.

The MAT1-2-1 protein sequences derived from 183 *H. sinensis* strains show 98.0–100% homology; 64 strains contain one to five conserved and/or nonconserved mutated residues out of 249 aa residues (Figure 3 and Appendix A) [10,36,37,38,39,59]. The MAT1-2-1 protein sequence translated from the metatranscriptome of OSIN7649 obtained from natural *C. sinensis* collected from Deqin County in Yunnan Province shares 100% identity with the authentic MAT1-2-1 protein sequence AFX66389 derived from the genome sequence KC429550 of the *H. sinensis* strain GS09_121 [54,59]. However, the MAT1-2-1 transcript was absent in the metatranscriptome assembly GAGW00000000 of natural *C. sinensis* collected from Kangding County in Sichuan Province (*cf*. Table 2) [53]. Table 3 shows no change in the amino acid components of the MAT1-2-1 proteins AFX66389 of the *H. sinensis* strains GS09_121 and OSIN7649 of natural *C. sinensis*, indicating that the hydrophobicity of the proteins was unchanged [54,59].

In addition to the above bioinformatic analyses of protein sequences, Zhang and Zhang [39] reported 4.7% and 5.7% allelic variations in the coding sequences of the MAT1-1-1 and MAT1-2-1 genes, respectively, resulting in 5.9% and 5.6% variations in the MAT1-1-1 and MAT1-2-1 protein sequences, respectively. The allelic and amino acid variations are much greater than what we observed through GenBank database analysis, suggesting that Zhang and Zhang [39] may not have uploaded all the gene and protein sequences that they analyzed to GenBank. It is conceivable that the mutant MAT1-1-1 and MAT1-2-1 proteins of natural *C. sinensis* might not necessarily be produced by Genotype #1 *H. sinensis* of *O. sinensis* but might be produced by one of multiple genotypes of *O. sinensis* fungi and co-colonized fungi when considering the failure to detect mating-type genes and transcripts within the *MAT1-1* and *MAT1-2* idiomorphs in numerous *H. sinensis* strains or wild-type isolates from natural *C. sinensis* (*cf*. Table 2 and Appendix A) and the failure “to induce development of the *C. sinensis* fruiting bodies” during 40 years of experience using pure *H. sinensis* cultures in research-oriented academic settings, as summarized by Zhang et al. [49].

### 3.6. Occurrence and Transcription of Pheromone Receptor Genes and Variations in Pheromone Receptor Proteins in H. sinensis and Natural C. sinensis

Heterothallic sexual development in morphologically indistinguishable fungal haploid cells is controlled by the growth of complementary mating mycelia that produce complementary mating proteins, followed by gametangial contact and the fusion of hyphal or ascosporic cells to induce plasmogamy and karyogamy in fungi [33,34,35]. These mating processes for ascomycetes occur between two types of fungal cells, a and α cells, and the mating of these fungal cells is regulated by secreted pheromones. Pheromones play essential roles in choreographing the interactions between mating partners by regulating intercellular mating communication and conveying information from one cell type to another cell type of the same species for physiological heterothallism or from a different species for hybridization. However, the biochemical processes of a- and α-pheromone precursors in ascomycetes appear to be complex and are not fully understood for *O. sinensis*. To date, scientists have not identified the genes encoding a- and α-pheromones in the five genome assemblies of *H. sinensis* strains 1229, CC1406-203, Co18, IOZ07, and ZJB12195 [36,43,44,45,46].

An α-pheromone is recognized by a-pheromone receptor (class 4 G-protein-coupled receptor) on the surface of an a-cell and vice versa [33,34,35]. The activated receptor plays crucial roles in the reciprocal interaction of mate recognition and the initiation of signaling and mating processes.

#### 3.6.1. a-Factor-like Pheromone Receptor in *H. sinensis* and Natural *C. sinensis*

Hu et al. [36] reported the occurrence of an a-pheromone receptor gene (*PRE-1*; 3→1386 of KE659055) in the *H. sinensis* strain Co18. It contains 3 exons separated by introns I and II (437→490 and 1018→1075 of KE659055, respectively) (Figure 4). The *PRE-1* gene sequences of the genome assemblies LKHE01002330 (1592→2975), NGJJ01000446 (84,292←85,675), JAAVMX010000007 (3,515,715→3,517,197), and ANOV01020521 (3→1386) of the *H. sinensis* strains 1229, CC1406-203, IOZ07, and Co18, respectively, show 100% homology with each other [36,43,45,46]. However, the sequence of the *PRE-1* gene exhibited 95.3% similarity with the genome assembly LWBQ01000004 (451,200→452,640) of the *H. sinensis* strain ZJB12195 due to multiple insertions and several transversion alleles in exon I (Figure 4, Table 4) [36,43,44,45,46]. The coding sequence of *PRE-1* is absent in the transcriptome assembly GCQL00000000 of the *H. sinensis* strain L0106 [52], indicating either silent transcription or the complete absence of the gene in a-cells in the *H. sinensis* strain L0106, resulting in impaired mate recognition upon interaction with the α-factor pheromone.

The *PRE-1* gene sequence is present in the metatranscriptome assemblies OSIN6252 and GAGW01004735/GAGW01004736 of natural *C. sinensis* specimens collected from Deqin County of Yunnan Province and Kangding County of Sichuan Province in China, respectively, showing 98.5–100% homology to the coding sequence of the *PRE-1* gene KE659055 of the *H. sinensis* strain Co18 (Table 4) [36,53,54].

The metatranscript OSIN6252 (220→1461) is 100% identical to the a-pheromone receptor gene sequence KE659055 of the *H. sinensis* strain Co18 (Figure 4). The metatranscript GAGW01004735 (1→764) is missing a 232-bp segment in exon I at its 5′ end and a 257-bp segment in exon III at its 3′ end, whereas the metatranscript GAGW01004736 (1029←1165) covers the missing portion of exon III at the 3′ end of the a-pheromone receptor gene sequence (Figure 4). Although the fragmentation of GAGW01004735 and GAGW01004736 may represent an assembly error if they are assumed to be derived from the same fungal genome [53], the integration of the two fragments is still 232 bp short at the 5′ end of exon I of the *PRE-1* gene.

In addition to the complete absence of the *PRE-1* transcript and protein in the *H. sinensis* strain L0106 [52], the protein sequences translated from the metatranscriptome assembly OSIN6252 of natural *C. sinensis* are 100% identical to the protein sequence EQK97482 of the a-pheromone receptor that was directly translated from the *PRE-1* gene KE659055 of the *H. sinensis* strain Co18 (Table 4; Figure 4 and Figure 5) [36,54]. The metatranscriptome GAGW01004735/GAGW01004736 of natural *C. sinensis* is 99.4% homologous to the protein sequence EQK97482 of the a-pheromone receptor (Table 4) [36,53]. The translated *PRE-1* protein sequence of GAGW01004735 is missing 75 aa and 85 aa at its N- and C-termini, respectively, whereas the other metatranscript, GAGW01004736, covers the C-terminal portion of the *PRE-1* protein with two nonconserved mutated aa residues (Figure 5).

Table 5 lists the four types of amino acids in a-pheromone receptor proteins on the basis of their chemical-physical and hydrophobic properties [68,69]. The segment deletion and nonconserved aa mutations in the *PRE-1* protein sequence produce a truncated a-pheromone receptor protein without significant changes in hydrophobicity (50.4–50.7% hydrophobic amino acids) (Table 5) or in the α-helix, β-sheet, β-turn, and coil plots of the a-pheromone receptor proteins (GAGW01004735-GAGW01004736 and OSIN6252) in natural *C. sinensis* [53,54], which raises the question of whether the mutations cause altered conformation of the membrane proteins and their functionality in interindividual mating signal communication between mating partners.

#### 3.6.2. α-Factor-like Pheromone Receptor in *H. sinensis* and Natural *C. sinensis*

Hu et al. [36] reported the occurrence of the α-pheromone receptor gene (2275→3724 of KE653642) in the *H. sinensis* strain Co18. This receptor gene shares 97.5–100% homology with the genome assemblies LKHE01001069 (13,389→13,603)/LKHE01000580 (23,846←24,749; 23,657←23,794), NGJJ01001310 (128,759←130,010), ANOV01006352 (1→138)/ANOV01006351 (2275→3528), JAAVMX010000012 (406,964→408,405) and LWBQ01000044 (488,856→490,264) in the *H. sinensis* strains 1229, CC1406-203, Co18, IOZ07 and ZJB12195, respectively (Figure 6, Table 4) [36,43,44,45,46]. Introns I and II of the α-pheromone receptor gene are located at sites 3091→3240 and 3528→3586 of the sequence KE653642 of strain Co18 [36]. A 133-bp segment deletion in exon I of the gene occurs between the genome assembly segments of LKHE01001069 and LKHE01000580 of the *H. sinensis* strain 1229 (Figure 6), which encodes a 45-aa deleted segment (ALVVAFAVLALTPAAKLRRPSSLLHLAGLAMCLARVGSLAVPALS) in the protein sequence.

The α-pheromone receptor gene contains three exons and two introns (Figure 6). The coding sequence of the gene of the *H. sinensis* strain Co18 shows 99.3–99.7% homology to the transcriptome assemblies GCQL01007648/GCQL01017756/GCQL01015779 of the *H. sinensis* strain L0106 and the metatranscriptome assembly OSIN6424 of natural *C. sinensis* collected from Deqin County of Yunnan Province, China (Table 3) [36,52,54]. GCQL01007648 (1←235) covers only 235 bp at the 5′ end of the 720 bp exon I. GCQL01015779 (1→908) and GCQL01017756 (1274←2180) cover 444 bp and 436 bp, respectively, at the 3′ end of exon I and the full-length sequences of exons II and III. There are two gaps (138 bp and 145 bp) between the sequences of GCQL01007648 and GCQL01017756 and between those of GCQL01007648 and GCQL01015779 [52].

The presence of the α-pheromone receptor transcript but not the a-pheromone receptor transcript in the *H. sinensis* strain L0106 suggests that the strain can only receive a mating signal from a sexual partner that contains a-cells and produces a-pheromone. The differential transcription of pheromone receptor genes in the *H. sinensis* strain L0106 (Table 4), together with the differential transcription of mating-type genes of the *MAT1-1* and *MAT1-2* idiomorphs (*cf*. Table 2), indicates that the *H. sinensis* strain L0106 needs a sexual partner that produces the a-pheromone and protein(s) of the *MAT1-1* idiomorph for sexual reproduction. The sexual partner may be the same species for physiological heterothallism or a different species for hybridization when homothallic or pseudohomothallic reproduction is impossible, as described above.

The metatranscriptome assembly OSIN6424 of natural *C. sinensis* contains the sequences of all 3 exons of the α-pheromone receptor gene with a 58-bp deletion (GTCGTCATCATCCTCCCCCTGGGCACCCTCGCCGCCCAGCCATGACAGCCCTCC-CA) between nucleotides 885 and 886 in exon II of the cDNA (Figure 6) [54]. Notably, the 58 bp deletion is not a multiple of 3 and may represent a frameshift mutation that involves a peptide segment of 52 aa encoded by the 3′ end region of exon II with no stop codon, whereas intron II of the gene is likely in “Intron Phase 0”, and the frameshift mutation may not involve exon III of the gene.

The α-pheromone receptor gene is absent in the metatranscriptome assembly GAGW00000000 of natural *C. sinensis* collected from Kangding County, Sichuan Province [53].

The translated protein sequence of the metatranscriptome assembly OSIN6424 (1→1224) from natural *C. sinensis* shows 82.9% similarity to the α-pheromone receptor protein sequence EQK99119, which was translated directly from the segment sequence of KE653642 of the *H. sinensis* strain Co18. OSIN6424 contains an insertion (38 aa), two segment deletions (9 and 11 aa), and 15-aa residues with nonconserved mutations (Figure 7, Table 4) [36,54].

Compared with the percentages of the 4 types of amino acids of the α-pheromone receptor protein EQK99119 of the *H. sinensis* strain Co18, as shown in Table 5, changes in the amino acids of the α-pheromone receptor protein sequences translated from the transcriptome assemblies GCQL01017756 and GCQL01015779 of the *H. sinensis* strain L0106 and the metatranscriptome assembly OSIN6424 of natural *C. sinensis* indicated dramatic decreases in hydrophobic amino acids and complementary increases in acidic, basic, and neutral amino acids [36,52,54].

Figure 8 shows the ExPASy ProtScale plots for the α-helices (Panel 8A), β-sheets (Panel 8B), β-turns (Panel 8C), and coils (Panel 8D) of the α-pheromone receptor proteins: the authentic protein EQK99119 of the *H. sinensis* strain Co18 and the variable α-pheromone receptor sequences translated from the transcriptome assemblies GCQL01017756 and GCQL01015779 of the *H. sinensis* strain L0106 and the metatranscriptome assembly OSIN6424 of natural *C. sinensis*. Substantial waveform changes were found in all the GCQL01017756, GCQL01015779, and OSIN6424 plots and are outlined with the open boxes in green.

Zheng & Wang [34] reported that the α-pheromone receptor is a G protein-coupled receptor possessing seven transmembrane domains. The membrane protein of the α-pheromone receptor EQK99119 of the *H. sinensis* strain Co18 contains 55.1% hydrophobic amino acids (Table 5), which is significantly greater than the approximately 41% hydrophobic amino acids in the MAT1-1-1 proteins (Table 3). The mutant α-pheromone receptor proteins in *H. sinensis* strain L0106 and natural *C. sinensis* contain fewer hydrophobic amino acids (50.8–52.7%) (Table 5). The alterations in the hydrophobicity (*cf*. Table 5) and the α-helix, β-sheet, β-turn, and coil plots (*cf*. Figure 8) of the α-pheromone receptor proteins of *H. sinensis* strain L0106 and natural *C. sinensis* indicate variable folding and secondary structures of the receptor proteins, indicating altered functionality of the sexual signal reception from mating partners and, in turn, the sexual reproduction of *O. sinensis,* probably favoring hybridization.

### 3.7. Other Pheromone-Related Genes in H. sinensis and Natural C. sinensis

Hu et al. [36] also identified genes encoding a pheromone-regulated membrane protein (65,257→66,887 of KE652182), the a-pheromone processing metallopeptidase Ste23 (10361→13752 of KE652396), and other related genes in the *H. sinensis* strain Co18. The proteins encoded by these genes participate in the biological processes of mating protein activation and signal transduction. The segment sequences of KE652182 and KE652396 exhibit 99.9–100% homology to the genome assemblies LKHE01001238 (91,967→93,597)/LKHE01000585 (20,198←23,589), NGJJ01000102 (63,761→65,391)/NGJJ01001093 (42,301→45,691), ANOV01000086 (17,505→19,135)/ANOV01001222 (10,361→13,752) and JAAVMX010000009 (1,427,620→1,429,250)/JAAVMX010000008 (1,724,511→1,727,903) of the *H. sinensis* strains 1229, CC1406-203, Co18 and IOZ07, respectively, but 95.8–98.4% similarity to the genome assemblies LWBQ01000047 (64,210→65,902)/LWBQ01000028 (337,499←339,681) of strain ZJB12195 due to multiple insertion/deletion mutations and other transition and transversion mutations in alleles [36,43,44,45,46].

The KE652182 gene encoding a pheromone-regulated membrane protein shows 95.2–96.3% similarity to the transcriptome assembly GCQL01011718 of the *H. sinensis* strain L0106 and the metatranscriptome assemblies GAGW01006658 and OSIN1278 of natural *C. sinensis* collected from Kangding County of Sichuan Province and Deqin County of Yunnan Province, respectively, with five to six deletion gaps [36,52,53,54].

The KE652396 gene encoding the a-pheromone processing metallopeptidase Ste23 is 99.2–100% homologous to the transcriptome assembly GCQL010123128 of the *H. sinensis* strain L0106 and the metatranscriptome assemblies GAGW01006419/GAGW01005794 and OSIN0988 of natural *C. sinensis* collected from Kangding County of Sichuan Province and Deqin County of Yunnan Province, respectively [36,52,53,54].

## 4. Discussion

### 4.1. Reproductive Behavior of H. sinensis, Genotype #1 of O. sinensis

Three reproduction hypotheses have been previously proposed for *H. sinensis*, the postulated sole anamorph of *O. sinensis* [5,6]: homothallism [36], pseudohomothallism [37], and facultative hybridization [39]. In theory, self-fertilization in ascomycetes becomes a reality when the mating-type genes of both the *MAT1-1* and *MAT1-2* idiomorphs are successfully transcribed and translated and when the mating proteins are synthesized and fully activated within a single fungal cell [32,33,34,35,36,37,38,39,59]. According to the bioinformatic findings presented in this paper, the differential occurrence of the mating-type genes of the *MAT1-1* and *MAT1-2* idiomorphs in 237 wild *C. sinensis* isolates and *H. sinensis* strains (*cf*. Appendix A) fails to support the genetic-based capability of self-fertilization. The transcriptome assembly GCQL00000000 of the *H. sinensis* strain L0106 contains a transcript of the MAT1-2-1 gene but no transcripts of the *MAT1-1* idiomorph [52]. The differential occurrence and transcription of mating-type genes at the genomic and transcriptomic levels are not consistent with the self-fertilization hypothesis for *H. sinensis* under homothallic or pseudohomothallic reproduction; instead, they support a reproductive strategy of either physiological heterothallism or hybridization [34,35,60,63,70,71,72,73].

Bushley et al. [37] detected the full sequences of the MAT1-1-1, MAT1-1-2, MAT1-1-3, and MAT1-2-1 genes in single-ascospore isolates via a genome walking/tail PCR strategy. They also observed multicellular heterokaryotic hyphae and ascospores of natural *C. sinensis* with mononucleated, binucleated, trinucleated, and tetranucleated structures (*cf*. Figure 3 of [37]). However, they detected the alternatively spliced transcript of the MAT1-2-1 gene of the *H. sinensis* strain 1229 with an unspliced intron I that contains 3 stop codons (*cf*. Appendix A); possible technical errors of DNA contamination were ruled out because intron II was normally spliced. This transcription phenomenon indicates translational interruption of the MAT1-2-1 transcript, and the production of a largely truncated MAT1-2-1 protein encoded only by exon I of the gene without the majority of the protein, which is encoded by exons II and III. This inability to produce a full-length and functional MAT1-2-1 protein resulting in dysfunctional MAT1-2-1 mating might constitute a mechanism of coupled transcriptional-translational control of *O. sinensis* sexual reproduction.

Li et al. [13] obtained 15 cultures from mono-ascospores of natural *C. sinensis*: 7 homogeneous clones contained only *H. sinensis* (GC-biased Genotype #1 of *O. sinensis*), and 8 other clones heterogeneously contained both the GC-biased Genotype #1 and AT-biased Genotype #5 of *O. sinensis*. The GC- and AT-biased sequences of *O. sinensis* reside in independent genomes and belong to different fungi [5,6,7,8,9,11]. Li et al. [14,15] observed two types of ascospores (fully and semiejected) of natural *C. sinensis* and reported the coexistence of the GC-biased Genotypes #1 and #13–14 of *O. sinensis*, AT-biased Genotypes #5–6 and #16 of *O. sinensis*, *Paecilomyces hepiali* and an AB067719-type fungus. Zhang and Zhang [39] hypothesized that the nuclei of binucleated hyphal and ascosporic cells (as well as mononucleated, trinucleated, and tetranucleated cells) of natural *C. sinensis* possibly contain different genetic material in two or more sets of genomes of independent fungal species, which might produce complementary mating-type proteins for sexual reproductive outcrossing.

In addition to the translated protein sequences derived from the genome assembly sequences of five *H. sinensis* strains, GenBank also lists 183 MAT1-2-1 protein sequences of various *H. sinensis* strains, which were not obtained directly through protein purification and amino acid sequencing but instead derived from the nucleotide sequences of the MAT1-2-1 gene or transcript of *H. sinensis* [10,36,37,38,39,59]. These protein sequences show 98.4–100% similarity to the translated protein sequence of the transcriptome assembly GCQL01020543 of strain L0106 [52]. In comparison, Zhang and Zhang [39] reported much greater allelic variations (4.7% and 5.7%) in the coding sequences of the *H. sinensis* MAT1-1-1 and MAT1-2-1 genes, respectively, predicting 5.9% and 5.6% variations in the amino acid sequences of the coding sequences and possible translation disturbances.

As mentioned above, the MAT1-1-1 gene is not expressed in the *H. sinensis* strains CS2, L0106, and SCK05-4-3 and is even absent in the genomes of many other *H. sinensis* strains (*cf*. Appendix A and Table 2) [10,38,39,52,59]. In addition, the alternatively spliced transcript of the MAT1-2-1 gene with an unspliced intron I and spliced intron II in the *H. sinensis* strain 1229 provides a translation template for producing a largely truncated and dysfunctional MAT1-2-1 protein encoded by exon I but not by exons II and III, which constitutes an example of coupled transcriptional-translational regulation of the mating process [37]. To date, there have been no experimental reports on the parallel production and direct amino acid sequencing of the mating proteins of both the *MAT1-1* and *MAT1-2* idiomorphs within a pure culture of *H. sinensis*, although many papers have used the word “expression” to describe the transcription of mating-type genes without considering other aspects of gene expression, such as epigenetic, posttranscriptional, translational, and posttranslational modifications and protein activation/degradation processes.

On the basis of the differential occurrence and transcription of the mating-type genes, regardless of whether *H. sinensis* (the postulated sole anamorph of *O. sinensis*) is monoecious or dioecious, there may be two or more *H. sinensis* populations capable of producing either mating proteins of the *MAT1-1* and *MAT1-2* idiomorphs and functioning reciprocally as sexual partners for successful physiological heterothallism crossing. If this assumption is correct, the sexual partners might possess indistinguishable *H. sinensis*-like morphological and growth characteristics [7,10,12,13,16,41]. For example, the indistinguishable *H. sinensis* strains 1229 and L0106 produce complementary transcripts of the mating-type genes and mating proteins of the *MAT1-1* and *MAT1-2* idiomorphs, as well as the a- and α-pheromone receptor genes, which are differentially transcribed and produce variable receptor proteins in the indistinguishable *H. sinensis* strains Co18 and L0106. If the physiological heterothallism hypothesis is incorrect for *O. sinensis*, one of the mating proteins might be produced by heterospecific fungal species, which would result in plasmogamy and the formation of heterokaryotic cells (*cf*. Figure 3 of [37]) to ensure a successful hybridization process if the heterospecific species are able to break interspecific reproduction isolation, similar to many cases of fungal hybridization that probably facilitate adaptation to the extremely adverse ecological environment on the Tibet–Qinghai Plateau [70,71,72,73,74]. Alternatively, to complete physiological heterothallism or hybridization reproduction, mating partners might exist in three-dimensionally adjacent hyphal cells, which might make their mating choices and communicate with each other through a mating signal-based transduction system of pheromones and pheromone receptors and form “H”-shaped crossings of multicellular hyphae, as observed by Hu et al. [36], Bushley et al. [37], and Mao et al. [16]. In particular, Mao et al. [16] reported the observation of “H”-shaped morphology in *C. sinensis* hyphae that contained either AT-biased Genotype #4 or #5 of *O. sinensis* without the co-occurrence of the GC-biased Genotype #1 *H. sinensis,* and the AT-biased *O. sinensis* genotypes shared indistinguishable *H. sinensis*-like morphological and growth characteristics.

### 4.2. Sexual Reproduction Strategy during the Lifecycle of Natural C. sinensis

The differential occurrence and transcription of the mating-type genes of both the *MAT1-1* and *MAT1-2* idiomorphs have also been observed in natural and cultivated *C. sinensis*, which contain multiple genotypes of *O. sinensis* and numerous fungal species [5,6,7,9,10,11,12,13,14,15,16,41,47,63,64,65,66,67]. As described above, transcriptome and metatranscriptome studies have shown differential transcription of mating-type genes in different maturation stages of natural and cultivated *C. sinensis* (*cf*. Appendix A), with a wide range of similarities compared with the sequences of the *H. sinensis* genes and transcripts and the unassembled metatranscriptome sequence reads, possibly indicating heterogeneous fungal sources of the transcripts. The middle and N-terminal truncated MAT1-1-1 proteins observed in natural *C. sinensis* exhibit variable hydrophobicity and alterations in the α-helices, β-sheets, β-turns, and coils (*cf*. Figure 1, Figure 2 and Appendix A, Table 3), suggesting heteromorphic folding and altered primary and secondary structures of the MAT1-1-1 proteins, which could result in dysfunctional or anomalous fungal mating processes and may indicate the heterospecific fungal sources of the proteins needed for hybridization in natural *C. sinensis*. In addition, the variable α-pheromone receptor proteins observed in natural *C. sinensis* exhibit changes in hydrophobicity and in the α-helices, β-sheets, β-turns, and coils of the proteins (*cf*. Figure 6, Figure 7 and Figure 8, Table 5), indicating altered primary and secondary structures of the α-pheromone receptor proteins that could result in altered functionality in the sexual signal reception from mating partners in natural *C. sinensis*. Although the coexistence of larval tissues and fungal mycelia in the caterpillar body of natural *C. sinensis* indicates that larval tissues are not just culture media that passively provide nutrients for fungal growth [5], the impact of host immunological reactions on the mating-type gene transcription of *O. sinensis* has not been explored, which represents the key process in the production of *O. sinensis* fruiting bodies and in the sexual life of natural and cultivated *C. sinensis*.

Regardless of whether *H. sinensis* (Genotype #1 of *O. sinensis*) is monoecious or dioecious, the sexual reproductive process of *O. sinensis* might require mating partners with the same or a different genotype of *O. sinensis* or even another fungal species to produce complementary mating proteins for physiological heterothallism or hybridization outcrossing. Thus, *O. sinensis* might have more than one anamorph to achieve sexual reproduction of *O. sinensis* and to accomplish the lifecycle of natural *C. sinensis*. These fungal partners might be located within a single heterokaryotic hyphal and ascosporic cell with mononucleated, binucleated, trinucleated, or tetranucleated structures (*cf*. Figure 3 of [37]), which would suggest that the following scientific observations from prior studies need to be reassessed:Li et al. [13] detected GC-biased *H. sinensis* (Genotype #1) and AT-biased Genotype #5 of *O. sinensis* in eight of 15 heterogeneous cultures (1206, 1208, 1209, 1214, 1220, 1224, 1227, and 1228) from mono-ascospores of natural *C. sinensis*, in addition to 7 other cultures that contained homogenous GC-biased *H. sinensis* (1207, 1218, 1219, 1221, 1222, 1225, and 1229). In addition to the nondisclosure of 9 other possible ascosporic clones (namely, clones 1210, 1211, 1212, 1213, 1215, 1216, 1217, 1223, and 1226), the authors misinterpreted all the AT-biased genotypes as the “ITS pseudogene” components of the *H. sinensis* genome, whereas AT-biased Genotypes #4, #6, and #15–17 were not detected by Li et al. [13] in the cultures of the mono-ascospores or the sequences of all the AT-biased genotypes residing not in the genomes of the GC-biased *H. sinensis* strains 1229, CC1406-203, Co18, IOZ07, and ZJB12195 but instead in the genomes of independent *O. sinensis* fungi [5,6,9,14,15,36,43,44,45,46]. These results suggest the possibility that GC-biased *H. sinensis* (Genotype #1) and AT-biased Genotype #5 of *O. sinensis* may become sexual partners to accomplish the sexual reproduction of *O. sinensis*.Zhu et al. [11] demonstrated the co-occurrence of GC-biased Genotypes #1 and #2 of *O. sinensis* in the stromata of natural *C. sinensis* during maturation. The ITS sequences of these two genotypes share 94.7% sequence similarity [5,6,15]. The Genotype #2 sequences were located outside the phylogenetic clade of Genotype #1 in the Bayesian trees (*cf*. Figure 7 of [15] and Figure 2 of [5]) and did not reside in the genome of Genotype #1 *H. sinensis* [5,15]. The abundances of the two genotypes undergo dynamic alterations in a disproportional, asynchronous manner in the stromata of natural *C. sinensis* during maturation [5], indicating the genomic independence of the two genotypes as evidence of independent *O. sinensis* fungi. These results suggest the possibility that GC-biased Genotypes #1 and #2 of *O. sinensis* may become sexual partners to accomplish the sexual reproduction of *O. sinensis*.Chen et al. [12] reported the detection of the Genotype #1 *H. sinensis* sequence AJ488255 from the caterpillar body of a natural *C. sinensis* specimen (#H1023) collected from Qinghai Province in China and the Genotype #7 sequence AJ488254 with multiple transversions and transition point mutations from the stroma of the same specimen [5,6,15]. The GC-biased Genotype #7 sequence is located within the phylogenetic clade of GC-biased Genotype #1 in the Bayesian trees but does not reside in the genome of *H. sinensis* (GC-biased Genotype #1 of *O. sinensis*) [5,15]. These results suggest the possibility that GC-biased Genotypes #1 and #7 of *O. sinensis* may become sexual partners to accomplish the sexual reproduction of *O. sinensis*.The co-occurrence of multiple GC- and AT-biased genotypes of *O. sinensis* in different combinations has been observed in the stroma, caterpillar body, ascocarps, and ascospores of natural *C. sinensis* [5,6,9,11,14,15]. The sequences of the *O. sinensis* genotypes do not reside in the genome of the GC-biased *H. sinensis* but instead belong to the genomes of independent fungi [5,6,9,11,14,15,36,43,44,45,46]. The abundances of the GC- and AT-biased genotypes of *O. sinensis* undergo dynamic alterations in a disproportional, asynchronous manner in the caterpillar bodies and stromata of *C. sinensis* during maturation, with a consistent predominance of the AT-biased genotypes of *O. sinensis*, not the GC-biased *H. sinensis*, in the stromata [5,6,11]. These results suggest the possibility that GC-biased *H. sinensis* (Genotype #1) and one of AT-biased genotypes of *O. sinensis* may become sexual partners to accomplish the sexual reproduction of *O. sinensis*.Mao et al. [16] identified AT-biased Genotype #4 or #5 of *O. sinensis* fungi without the co-occurrence of GC-biased *H. sinensis* in natural *C. sinensis* specimens collected from geographically remote production areas. They also reported that AT-biased mutant genotypes presented indistinguishable *H. sinensis*-like morphologic and growth characteristics and were able to form “H”-shaped hyphal crossings and anastomoses during germination, which are related to the sexual reproduction of *O. sinensis*. Similarly, Kinjo and Zang [7] reported the detection of AT-biased Genotype #4 or #5 of *O. sinensis* in several natural *C. sinensis* samples collected from remote production areas and of GC-biased Genotype #1 *H. sinensis* in other *C. sinensis* specimens collected from different production areas. These results suggest the possibility that different AT-biased genotypes of *O. sinensis* may become sexual partners without the participation of GC-biased Genotype #1 *H. sinensis* to accomplish the sexual reproduction of *O. sinensis*.Hu et al. [36] reported the use of a mixture of two pure *H. sinensis* strains, Co18 and QH195-240, to inoculate 40 larvae of Hepialidae sp. Fungal inoculation induced death and mummification of the larvae but failed to induce the development of fruiting bodies and ascospores, indicating biological separation of the larval death/mummification process and the fungal fruiting body development process. The authors cited two other studies [61,62] and reported that inoculation of ghost moth larvae of the Hepialidae family with pure *H. sinensis* consistently failed to produce fruiting bodies and ascospores. Zhang et al. [49] (coauthors of [36]) summarized 40 years of experience in artificial cultivation of *C. sinensis* and concluded that “it is very difficult in our laboratory to induce development of the *C. sinensis* fruiting bodies, either on culture medium or on insects.” Our findings presented in this paper provide evidence at the genetic, transcriptional, and protein levels to explain the self-sterility of *H. sinensis* and the failure of the inoculation experiments using pure cultures of *H. sinensis* as the sole inoculant.Wei et al. [63] reported a species contradiction between anamorphic inoculants (3 strains of the GC-biased Genotype #1 *H.* sinensis: 130508-KD-2B, 20110514, and H01-20140924-03) and the only teleomorph of the AT-biased Genotype #4 of *O. sinensis* in the fruiting body of cultivated *C. sinensis*. In addition, Figure 6 of [63] shows two phylogenetically distinct teleomorphs of *O. sinensis*: the AT-biased Genotype #4 of *O. sinensis* in cultivated *C. sinensis* and the GC-biased Genotype #1 in the natural *C. sinensis* specimen G3, which was used as the teleomorphic reference in the phylogenetic analysis. Because the sequences of the AT- and GC-biased genotypes of *O. sinensis* reside in independent genomes of different fungi [5,9,11,36,43,44,45,46], Wei et al. [63] demonstrated two distinct teleomorphs of *O. sinensis* and questioned the true causal fungus/fungi and anamorph-teleomorph connections of *O. sinensis* according to Koch’s postulates and the sole anamorph and sole teleomorph hypotheses proposed 10 years ago by the same group of key authors [40]. Under the self-sterility hypothesis for *H. sinensis* presented in this bioinformatic paper, the AT-biased genotypes of *O. sinensis* may play a possible hybridization role as the mating partner(s) of the GC-biased *H. sinensis* if they belong to different fungal species, “even at higher taxonomic levels (genera and family)”.*Tolypocladium sinense* in natural *C. sinensis* was first identified and reported by Li [17]. It was subsequently isolated from natural *C. sinensis* and characterized morphologically and genetically [20,21]. Engh [41] reported the molecular identification of the *Cordyceps-Tolypocladium* complex in natural *C. sinensis*. The “*Cordyceps*” sequence AJ786590 obtained by Engh [41] was published and uploaded to GenBank by Stensrud et al. [42] and phylogenetically clustered into AT-biased Group B (Genotype #4) of *O. sinensis*, along with other *C. sinensis* sequences, by Stensrud et al. [8]. Barseghyan et al. [24] performed a macro/micromycology study and concluded that *H. sinensis*, which is presumed to be psychrophilic, and *T. sinensis*, which is presumed to be mesophilic, are dual anamorphs of *O. sinensis*. Notably, the *O. sinensis* fungus, which has *H. sinensis*-like morphology and growth characteristics, was not genotyped molecularly in that study. According to the self-sterility hypothesis for *H. sinensis* presented in this bioinformatic paper, the close association of *T. sinense* with *H. sinensis* may help mycological physiologists plan future studies to explore the possibility of *O. sinensis* hybridization reproduction.Genotypes #13 (KT339190) and #14 (KT339178) of *O. sinensis* have been identified in either semiejected or fully ejected multicellular heterokaryotic ascospores, respectively, collected from the same specimen of natural *C. sinensis* [5,6,15]. The two genotypes feature precise reciprocal substitutions of large DNA segments due to chromosomal intertwining interactions and genetic material recombination between two parental fungi, Genotype #1 *H. sinensis* (Group A by Stensrud et al. [8]) and an AB067719-type Group E fungus (Appendix A) [5,6,15]. A pure culture of the AB067719-type fungus has not been obtained, and its taxonomic position is unclear. More than 900 sequences highly homologous to AB067719, including those of *Alternaria* sp., *Ascomycota* sp., *Aspergillus* sp., *Avena* sp., *Berberis* sp., *Colletotrichum* sp., *Cordyceps* sp., *Cyanonectria* sp., *Dikarya* sp., *Fusarium* sp., *Gibberella* sp., *Hypocreales* sp., *Juglans* sp., *Lachnum* sp., *Nectria* sp., *Nectriaceae* sp., *Neonectria* sp., and *Penicillium* sp., have been uploaded to GenBank [15]. Chromosomal intertwining and genetic material recombination may occur after plasmogamy and karyogamy of heterospecific parental fungi under sexual reproduction hybridization or parasexuality, which is characterized by the prevalence of heterokaryosis and results in concerted chromosome loss for transferring/substituting genetic materials without conventional meiosis [37,75,76,77]. The phenomena of precise vertical transfer and reciprocal substitution of genetic materials between the chromosomes of heterospecific parental fungi that occurred differently between the two types of ascospores collected from the same specimen of natural *C. sinensis* are distinct from the randomness and arbitrariness of horizontal environmental gene drift.*P. hepiali* was first isolated from natural *C. sinensis* by Dai et al. [19,20]. A close association between psychrophilic *O. sinensis* (GC- and AT-biased genotypes) and mesophilic *P. hepiali* has been found in the caterpillar body, stroma, and stromal fertile portion, which are densely covered with ascocarps and ascospores of natural *C. sinensis,* and even in the formation of a fungal complex in “pure” *H. sinensis* strains that were isolated from natural *C. sinensis* and provided as gifts by a distinguished mycology taxonomist [6,11,15,20,22]. Whether certain strains of these fungal species would select each other as sexual partners will depend on their mating choices for hybridization and their ability to break interspecific isolation barriers to adapt to extremely harsh ecological environments on the Qinghai-Tibet Plateau and the seasonal change from the extremely cold winter when *C. sinensis* is in its asexual growth phase to the spring and early summer when *C. sinensis* switches to the sexual reproduction phase [70,71,72,73,74]. According to the self-sterility hypothesis for *H. sinensis* presented in this bioinformatic paper, *P. hepiali* may possibly play a hybridization role as the mating partner(s) of the GC- and AT-biased genotypes of *O. sinensis*.

## 5. Conclusions

Bioinformatic analysis of genomic, transcriptomic, and protein sequences available in public databases revealed the differential transcription of the mating-type genes of the *MAT1-1* and *MAT1-2* idiomorphs and pheromone receptor genes of *H. sinensis*. The genomic and transcriptomic evidence is inconsistent with the self-fertilization hypothesis under homothallism and pseudohomothallism for *H. sinensis* but instead suggests self-sterility in *O. sinensis*, which utilizes physiological heterothallism or fungal hybridization strategies for sexual reproduction in natural *C. sinensis*. The *H. sinensis* strains L0106 and 1229 differentially transcribe the mating-type genes of the *MAT1-1* and *MAT1-2* idiomorphs and might become sexual partners for physiological heterothallic outcrossing. The *H. sinensis* strain L0106 possesses the transcript of an α-pheromone receptor gene but not an a-pheromone receptor gene, indicating its ability to receive a mating signal from an a-pheromone secreted by a-cells of another fungus, either the same or different fungal species, to accomplish heterothallism or hybridization outcrossing for the development and maturation of the fruiting body, ascocarps, and ascospores of natural *C. sinensis*. The mutations of MAT1-1-1 and α-pheromone receptor proteins observed in natural *C. sinensis* result in dramatically altered hydrophobicity properties and secondary protein structures, suggesting the heterogeneity of the fungal source(s) of the proteins in the natural *C. sinensis* insect-fungal complex and sexual reproduction of *O. sinensis* under heterothallism or hybridization.

## Figures and Tables

**Figure 1 biology-13-00632-f001:**
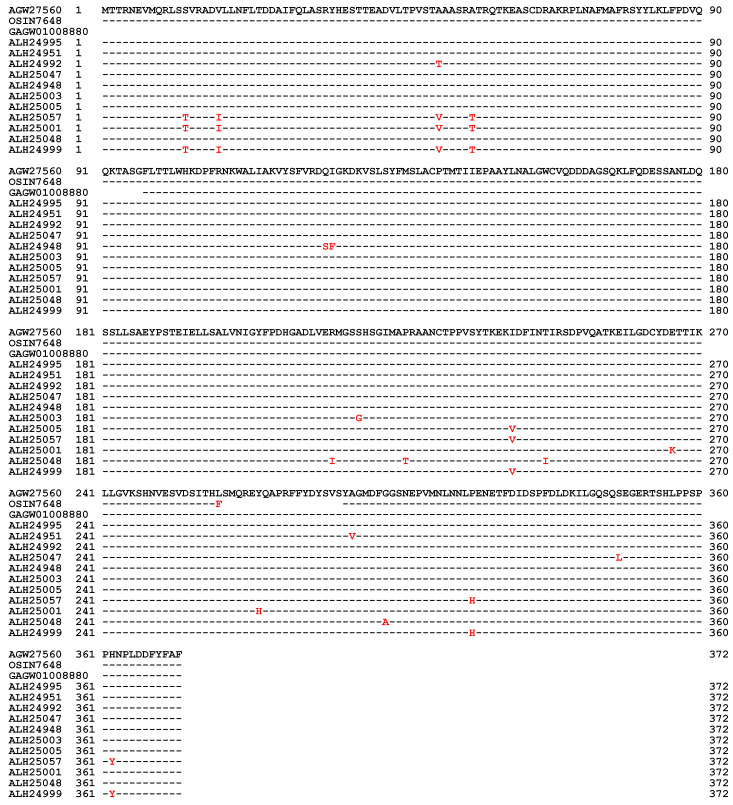
Alignment of the MAT1-1-1 protein sequences translated from the transcripts of *H. sinensis* strains and metatranscriptomes of natural *C. sinensis*. OSIN7648 and GAGW01008880 are MAT1-1-1 protein sequences derived from the metatranscriptome sequences of natural *C. sinensis* collected from Deqin County of Yunnan Province and Kangding County of Sichuan Province in China, respectively [53,54]. AGW27560, ALH24995, ALH24951, ALH24992, ALH25047, ALH24948, ALH25003, ALH25005, ALH25057, ALH25001, ALH25048, and ALH24999 are the MAT1-1-1 protein sequences of strains 1229, SC09_97, GS09_229, SC09_65, YN09_61, GS09_143, XZ05_6, XZ05_8, XZ12_16, XZ05_2, YN09_64, and XZ07_H2, respectively [37,38]. The residues in red indicate the variant amino acids. The hyphens indicate identical amino acid residues, and the spaces denote unmatched sequence gaps.

**Figure 2 biology-13-00632-f002:**
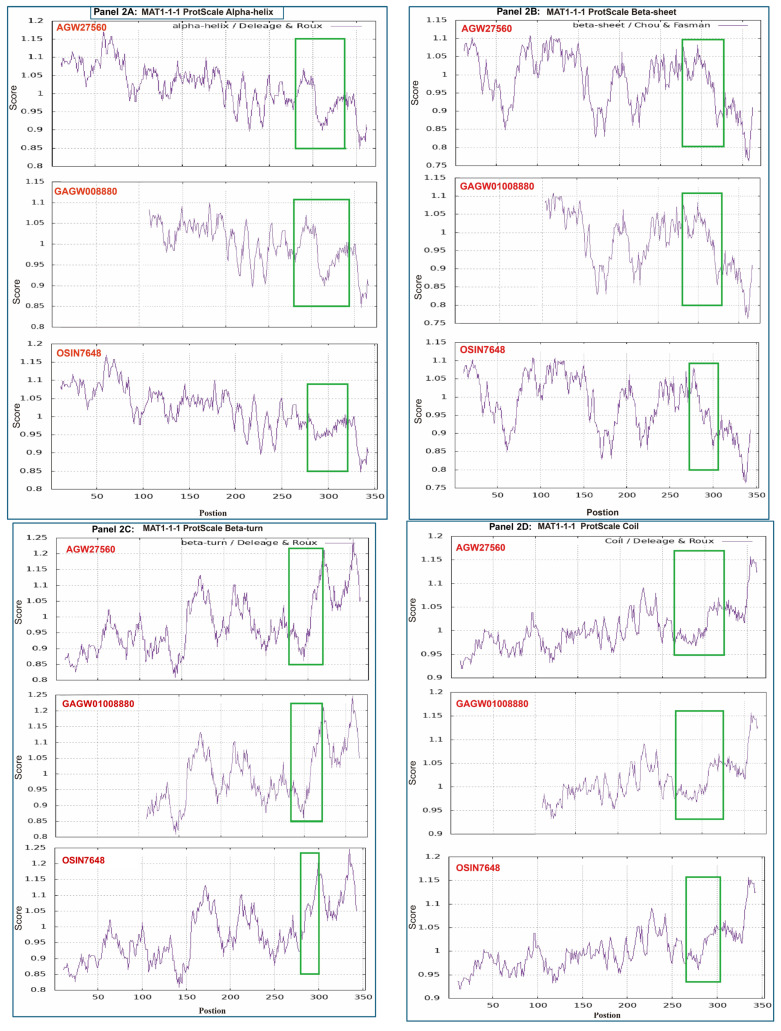
ExPASy ProtScale plots for the α-helices (**Panel 2A**), β-sheets (**Panel 2B**), β-turns **(Panel 2C**), and coils (**Panel 2D**) of the MAT1-1-1 proteins. Each panel contains three plots. The MAT1-1-1 protein sequence AGW27560 (372 aa; upper plots in all panels) of the *H. sinensis* strain CS68-2-1229 was compared with the metatranscriptome MAT1-1-1 sequences GAGW01008880 (276 aa; middle plots in all panels) and OSIN7648 (353 aa; lower plots in all panels) of natural *C. sinensis* (the original plots are included in Appendix A). The open boxes in green indicate the variable segment regions of the MAT1-1-1 proteins.

**Figure 3 biology-13-00632-f003:**
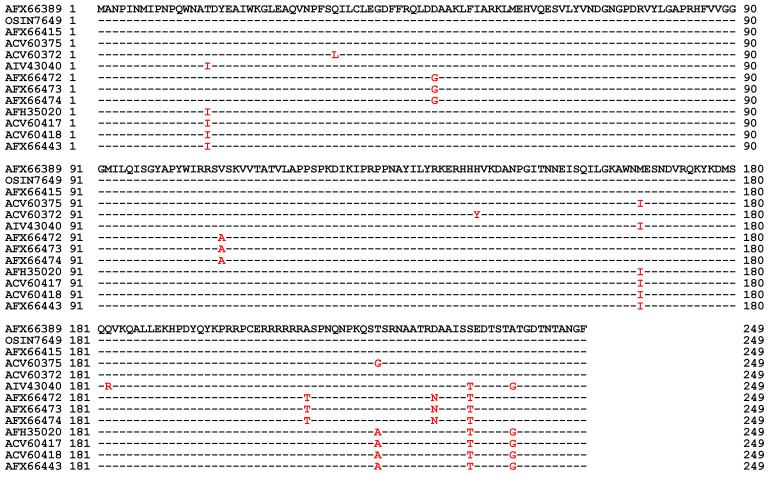
Alignment of the MAT1-2-1 protein sequences translated from the transcript sequences of *H. sinensis* strains and the metatranscriptome sequence of natural *C. sinensis*. OSIN7649 is the MAT1-2-1 protein sequence derived from the metatranscriptome sequence of natural *C. sinensis* collected from Deqin County, Yunnan Province, China [54]. AFX66389, AFX66415, ACV60375, ACV60372, AIV43040, AFX66472, AFX66473, AFX66474, AFH35020, ACV60417, ACV60418, and AFX66443 are the MAT1-2-1 protein sequences of the *H. sinensis* strains GS09_121, QH09_201, XZ-SN-44, XZ-LZ06-61, XZ12_16, YN09_6, YN09_22, YN09_51, XZ06-124, XZ-LZ07-H1, XZ-LZ07-H2, and XZ05_8, respectively [10,37,38,59]. The residues in red indicate the variant amino acids, and the hyphens indicate identical amino acid residues.

**Figure 4 biology-13-00632-f004:**
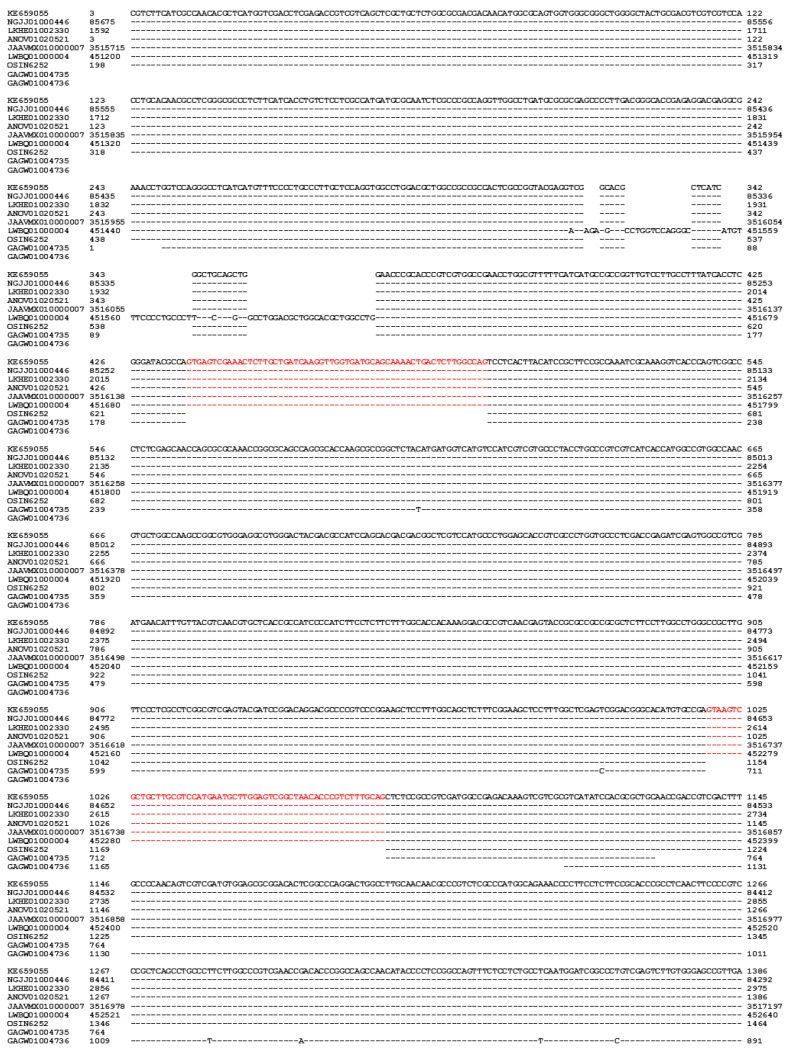
Alignment of the genome and transcriptome sequences of a-pheromone receptor (*PRE-1*) from *H. sinensis* strains and natural *C. sinensis*. KE659055 is the gene sequence of the *H. sinensis* strain Co18 [36]. The genome assemblies NGJJ01000446, LKHE01002330, ANOV01020521, JAAVMX010000007, and LWBQ01000004 were obtained from the *H. sinensis* strains CC1406-203, 1229, Co18, IOZ07, and ZJB12195, respectively [36,43,44,45,46]. OSIN6252 and GAGW01004735/GAGW01004736 are the transcriptome assembly sequences obtained from natural *C. sinensis* specimens collected from Deqin County of Yunnan Province and Kangding County of Sichuan Province of China, respectively [53,54]. The segments in red are introns I and II. The hyphens indicate identical bases, and the spaces denote unmatched sequence gaps.

**Figure 5 biology-13-00632-f005:**
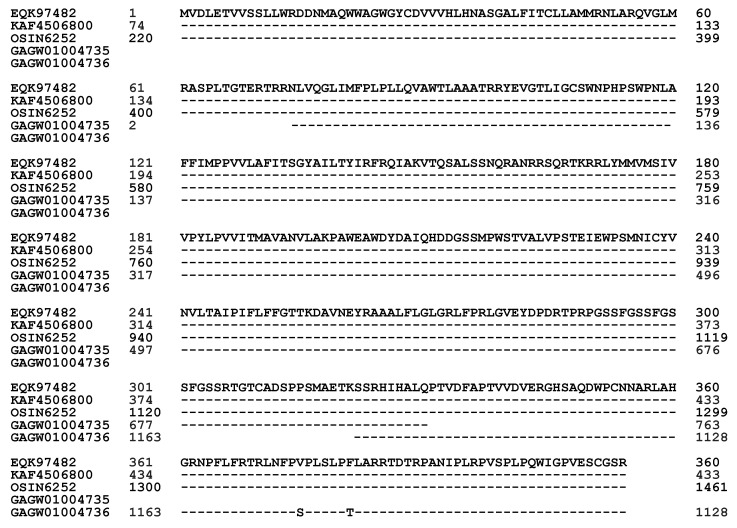
Alignment of the translated protein sequences of a-pheromone receptor (*PRE-1*) from *H. sinensis* strains and natural *C. sinensis*. The *PRE-1* protein sequences EQK97482 and KAF4506800 are translated from the gene sequence KE659055 of the *H. sinensis* strain Co18 and the genome segment sequence JAAVMX010000007 of the *H. sinensis* strain IOZ07 [36,46]. The *PRE-1* transcriptome assembly sequences OSIN6252, GAGW01004735, and GAGW01004736 of natural *C. sinensis* are translated into protein sequences [53,54]. The hyphens indicate identical bases, and the spaces denote unmatched sequence gaps.

**Figure 6 biology-13-00632-f006:**
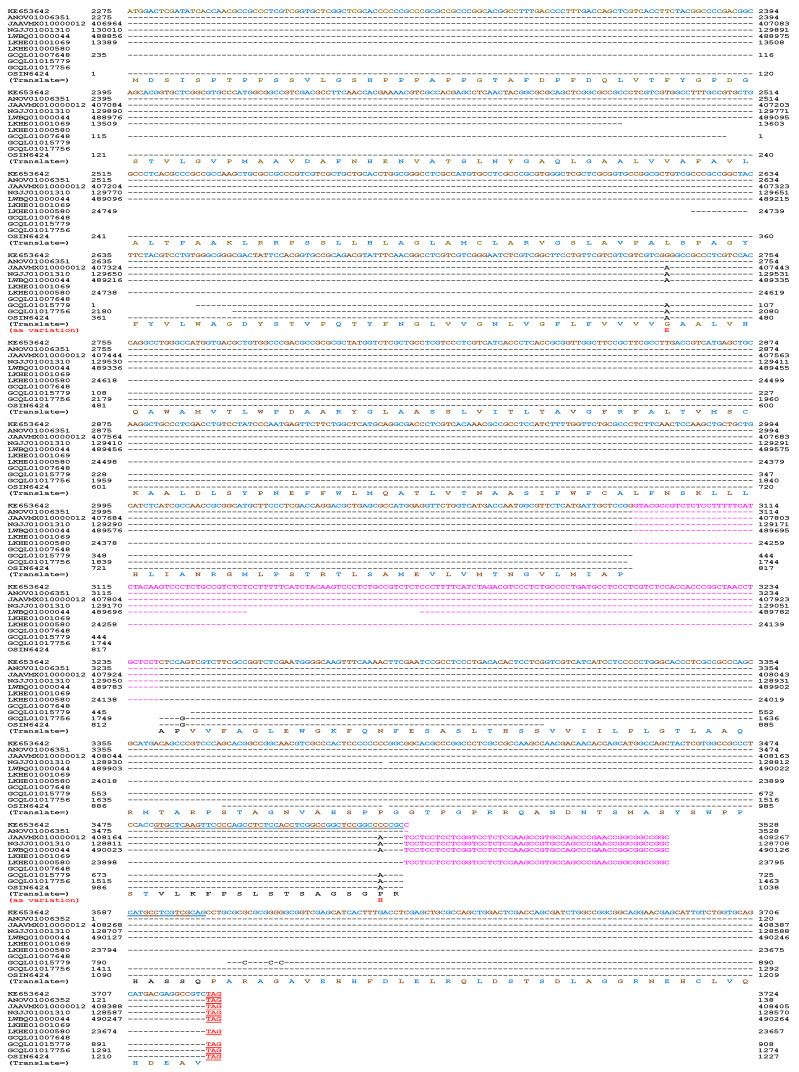
Alignment of the genome and transcriptome sequences of the α-pheromone receptor of *H. sinensis* and natural *C. sinensis*. The genome sequence of KE653642 was obtained from the *H. sinensis* strain Co18 [36]. JAAVMX010000012, NGJJ01001310, LWBQ0100044, ANOV01006352, and LKHE01000580 are the genome assemblies obtained from the *H. sinensis* strains IOZ07, CC1406-203, ZJB12195, Co18, and 1229, respectively [36,43,44,45,46]. GCQL01007648, GCQL01017756, GCQL01015779, and OSIN6424 are the transcriptome assemblies obtained from the *H. sinensis* strain L0106 and natural *C. sinensis* specimens collected from Deqin County of Yunnan Province, China [52,53]. The triplets shown in alternating brown and blue indicate the open reading frame. The sequences in pink indicate introns I and II between nucleotides 3092→3240 and 3528→3586, respectively, of the α-pheromone receptor gene KE653642, and the underlined “TAG” triplets shown in red are stop codons. The protein sequence was translated from the transcript sequences. The amino acid residues in red represent variants. The hyphens indicate identical bases, and the spaces denote unmatched sequence gaps.

**Figure 7 biology-13-00632-f007:**
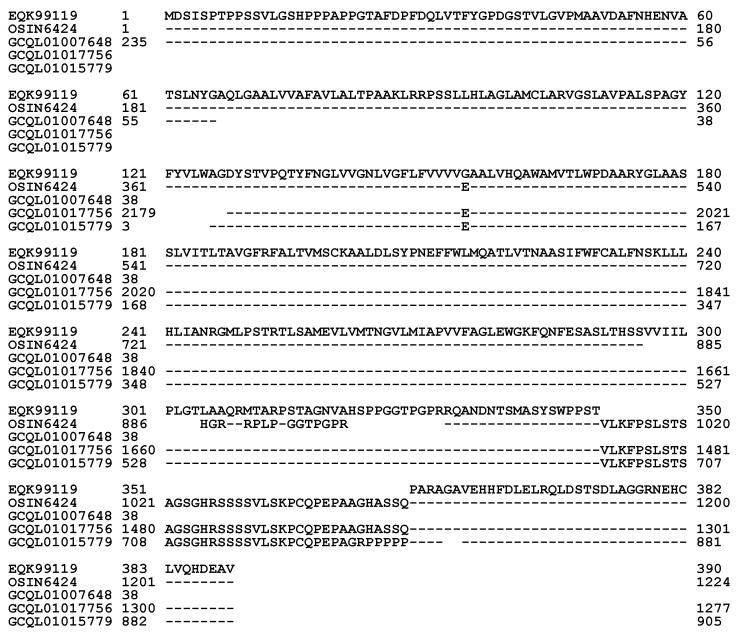
Alignment of the protein sequences of the α-pheromone receptor of the *H. sinensis* strain Co18 and the transcriptome sequences of the *H. sinensis* strain L0106 and natural *C. sinensis*. The gene sequence EQK99119 was obtained from the *H. sinensis* strain Co18 [36]. The transcriptome assemblies GCQL01007648, GCQL01017756, GCQL01015779, and OSIN6424 were obtained from the *H. sinensis* strain L0106 and the natural *C. sinensis* specimens collected from Deqin County, Yunnan Province, China [53,54]. The hyphens indicate identical amino acid residues, and the spaces denote unmatched sequence gaps.

**Figure 8 biology-13-00632-f008:**
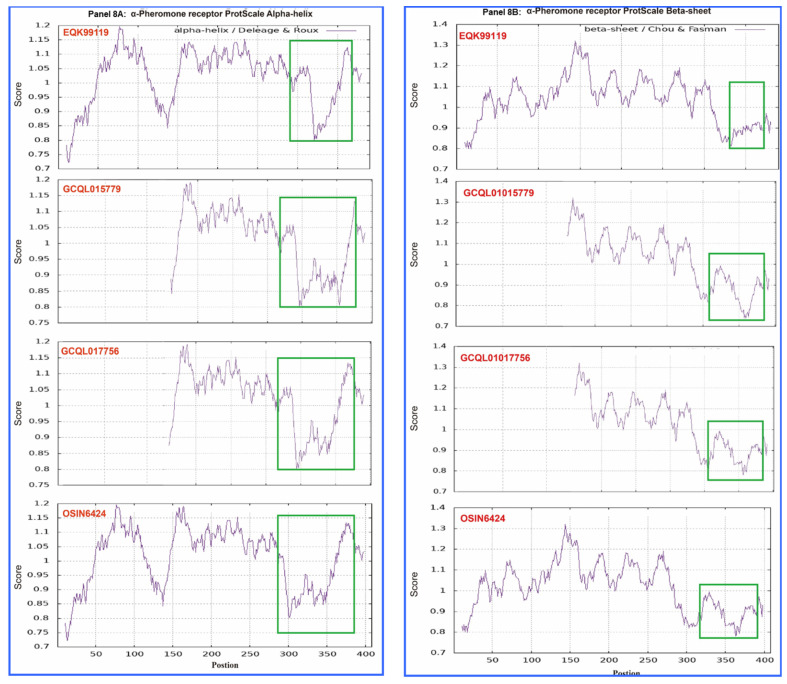
ExPASy ProtScale plots for the α-helices (**Panel 8A**), β-sheets (**Panel 8B**), β-turns (**Panel 8C**), and coils (**Panel 8D**) of α-pheromone receptor proteins. Each panel contains four plots. The α-pheromone receptor protein sequence EQK99119 (390 aa; upper plots in all panels) of the *H. sinensis* strain Co18 was compared with the transcriptome and metatranscriptome α-pheromone receptor sequences and GCQL01017756 (301 aa; upper middle plots in all panels) and GCQL01015779 (301 aa; lower middle plots in all panels) of the *H. sinensis* strain L0106 and OSIN6424 (408 aa; lower plots in all panels) of natural *C. sinensis* (the original pots are included in Appendix A). The open boxes in green indicate the variable segment regions of the α-pheromone receptor proteins.

**Table 1 biology-13-00632-t001:** Sequencing and assembly technologies (software) for genome and transcriptome assemblies of *H. sinensis* strains and natural *C. sinensis*.

GenBank Accession	*H. sinensis* Strain	Sequencing Method	Assembly Method	Ref.
ANOV00000000	Co18	Roche 454 GS FLX system (Illumina HiSeq: 454)	SOAPdenovo v.1.05 and Newbler v.2.3	[36]
JAAVMX000000000	IOZ07	PacBio Sequel sequencing technology	Canu v.1.7	[46]
LKHE00000000	1229	Illumina HiSeq sequencing technology	ABySS v.1.2.3	[43]
LWBQ00000000	ZJB12195	Illumina sequencing technology (HiSeq 2000 Sequencing System)	SOAPdenovo v.2.0	[44]
NGJJ00000000	CC1406-20395	Hierarchical Genome Assembly Process (HGAP) workflow (PacBioDevNet; Pacific Biosciences)	CA software (v.7.0) and the PacBio Rs_PreAssembler.1 module	[45]
GCQL00000000	L0106	Illumina HiSeq sequencing technology	SOAPdenovo v.2.0	[52]
GAGW00000000	Natural *C. sinensis* (Kangding, Sichuan, China)	454 technology	GS De Novo Assembler software v 2.6 or Newbler 2.6	[53]
(Metatranscriptome sequence not uploaded in GenBank)	Natural *C. sinensis* (Deqin, Yunnan, China)	Illumina HiSeq2000 platform	Trinity (version r20140717)	[54]

**Table 2 biology-13-00632-t002:** Percent similarity between the mating-type genes of the *MAT1-1* and *MAT1-2* idiomorphs and the genome and transcriptome sequences of *H. sinensis* and natural *C. sinensis*.

	MAT1-1-1	MAT1-1-2	MAT1-1-3	MAT1-2-1
	(vs. KC437356 of Strain CS68-2-1229)			(vs. JQ325153 of Strain GS09_121)
	% similarity of the genome sequences			
*H. sinensis * strain 1229	100%	100%	100%	99.9%
*H. sinensis * strain Co18	99.9%	100%	100%	99.7%
*H. sinensis * strain IOZ07	100%	100%	100%	―
*H. sinensis * strain CC1406-203	―	―	―	99.9%
*H. sinensis * strain ZJB12195	―	―	―	99.9%
	% similarity of the coding sequences of the genes (excluding introns) and the transcriptome sequences			
*H. sinensis * strain L0106	―	―	―	99.0%
*H. sinensis * strain 1229	100%	100%	100%	100%
Natural *C. sinensis* (Deqin, Yunnan)	100%	―	―	100%
Natural *C. sinensis* (Kangding, Sichuan)	100%	―	―	―

**Table 3 biology-13-00632-t003:** Percentages of amino acids in the MAT1-1-1 and MAT1-2-1 proteins of *H. sinensis* and natural *C. sinensis*.

		% Amino Acids of Mating-Type Protein			
		Hydrophobic	Acidic	Basic	Neutral
		MAT1-1-1 protein			
AGW27560	*H. sinensis * strain CS68-2-1229	41.4%	13.2%	11.0%	34.4%
GAGW01008880	Natural *C. sinensis*	41.3%	14.1%	9.8%	34.8%
OSIN7648	Natural *C. sinensis*	41.9%	13.1%	11.1%	33.7%
		MAT1-2-1 protein			
AFX66389	*H. sinensis * strain GS09_121	41.2%	10.0%	16.5%	33.3%
OSIN7649	Natural *C. sinensis*	41.2%	10.0%	16.5%	33.3%

**Table 4 biology-13-00632-t004:** Percentages of similarity between the protein sequences of pheromone receptors and the genome and transcriptome sequences of *H. sinensis* and natural *C. sinensis*.

	Percent Similarity	
	a-Factor-like Pheromone Receptor	α-Factor-like Pheromone Receptor
	Between the sequences of pheromone receptor genes and the *H. sinensis* genomes	
*H. sinensis * strain 1229	100%	99.8%
*H. sinensis * strain CC1406-203	100%	99.8%
*H. sinensis * strain Co18	100%	100%
*H. sinensis * strain IOZ07	100%	99.9%
*H. sinensis * strain ZJB12195	95.2%	97.5%
	Between the coding sequences of the pheromone receptor genes (excluding introns) and the transcriptome sequences of *H. sinensis* and natural *C. sinensis*	
*H. sinensis * strain L0106	―	99.3%
Natural *C. sinensis* (Deqin, Yunnan)	100%	99.7%
Natural *C. sinensis* (Kangding, Sichuan)	98.5–100%	―
	Between the pheromone receptor protein sequences translated from the gene sequences and the transcript sequences of *H. sinensis* and natural *C. sinensis*	
*H. sinensis * strain L0106	―	86.4–100%
Natural *C. sinensis* (Deqin, Yunnan)	100%	82.9%
Natural *C. sinensis* (Kangding, Sichuan)	97.8–100%	―

**Table 5 biology-13-00632-t005:** Percentages of amino acids in the pheromone receptor proteins of *H. sinensis* and natural *C. sinensis*.

		% Amino Acids of Pheromone Receptor Proteins			
		Hydrophobic	Acidic	Basic	Neutral
		a-pheromone receptor protein			
EQK97482	*H. sinensis * strain Co18	50.7%	6.5%	11.6%	31.4%
GAGW01004735-GAGW01004736	Natural *C. sinensis*	50.4%	6.2%	11.2%	31.2%
OSIN6252	Natural *C. sinensis*	50.7%	6.5%	11.6%	31.4%
		α-pheromone receptor protein			
EQK99119	*H. sinensis * strain Co18	55.1%	5.6%	7.4%	31.8%
GCQL01017756	*H. sinensis * strain L0106	50.8%	6.0%	9.0%	34.2%
GCQL01015779	*H. sinensis * strain L0106	51.8%	6.0%	9.0%	33.2%
OSIN6424	Natural *C. sinensis*	52.7%	5.8%	8.6%	32.8%

## Data Availability

Most of the gene, genome, transcript, transcriptome, metatranscriptome, and protein sequences are available in NCBI GenBank (http://www.ncbi.nlm.nih.gov/genbank/) (frequently accessed). The metatranscriptome sequences obtained from the natural *C. sinensis* specimens collected from Deqin, Yunnan Province, China, were submitted to www.plantkingdomgdb.com/Ophiocordyceps_sinensis/data/cds/Ophiocordyceps_sinensis_CDS.fas (accessed on 18 January 2018) [54]. However, this website is currently inaccessible, but a previously downloaded cDNA file was used for bioinformatic analysis. To obtain the sequence files, please send your inquiry to the original authors [54]: Xia, E.-H., Yang, D.-R., Jiang, J.-J., Zhang, Q.-J., Liu, Y., Liu, Y.-L., Zhang, Y., Zhang, H.-B., Shi, C., Tong, Y., Kim, C.-H., Chen, H., Peng, Y.-Q., Yu, Y., Zhang, W., Eichler, E.E., Gao, L.-Z. The caterpillar fungus, *Ophiocordyceps sinensis*, genome provides insights into highland adaptation of fungal pathogenicity. Sci. Rep. 2017; 7: 1806. DOI:10.1038/s41598-017-01869-z.

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
