# Peer review of "Mutations and Differential Transcription of Mating-Type and Pheromone Receptor Genes in Hirsutella sinensis and the Natural Cordyceps sinensis Insect-Fungi Complex"

_biology, 2024, doi:10.3390/biology13080632_

Round 1

Reviewer 1 Report (Previous Reviewer 1)

Comments and Suggestions for Authors

The Authors followed step-by-step my suggestions, agreeing or discussing with them. The explanations and changes introduced into the manuscript are satisfactory for me. 

The main concern was that the work is too large in volume. Now, the whole volume is reduced from 45 to 38 pages, mainly thanks to moving particular Tables and figures to supplementary files.  

Sincerely,

Reviewer

Reviewer 2 Report (Previous Reviewer 3)

Comments and Suggestions for Authors

The paper is ready for publication.

Reviewer 3 Report (New Reviewer)

Comments and Suggestions for Authors

Thank you for considering me to review the manuscript titled: Mutations and differential transcription of mating-type and pheromone receptor genes in pure cultures of Hirsutella sinensis and natural Cordyceps sinensis. The manuscript provides a reasonably robust dataset and appears suitable for acceptance after minor revisions.

Suggestions:

Ensure thorough English language editing throughout the manuscript to rectify grammatical errors and shorten lengthy sentences.

Please enhance the abstract by providing more descriptive details regarding methodology and emphasizing the results obtained.

The introduction should be summarized and improved by elaborating on the important aspects associated with the genetic occurrence of mating-type genes. Besides, clarifying the knowledge gap and hypothesis and extending the objectives.

Old citations throughout the manuscript should be updated (Li et al. 1988, Dai et al. 1989, Kinjo & Zang 2001; ……)

Variations in mating proteins should be numbered 3.5, not 3.4, and the following subtitles should be updated.

Ensure consistent use of journal abbreviations and proper citation formatting throughout the manuscript. Some journals are abbreviated as "Intl. J. Med. Mushrooms.", while others are not “Protein Engineering, Design and Selection”. Please review and standardize the reference section according to style guidelines. Following the journal style guide or citation requirements is vital for uniformity and accuracy. 

Comments on the Quality of English Language

Moderate editing of English language required

This manuscript is a resubmission of an earlier submission. The following is a list of the peer review reports and author responses from that submission.

Round 1

Reviewer 1 Report

Comments and Suggestions for Authors

The work is public data reanalysis, regarding in silico analyses of available genomes, transcriptomes, metatransciptomes  of Hirsutella sinensis and natural Cordyceps sinensis mating type loci and analysis of amino-acids structure of their pheromone receptors.

For sure it is profound and comprehensive work, but in my opinion too extensive material for one publication (only “Results” section has 30 pages!, …there are 95 reference positions). It is really huge work.

Other remarks:

In the title: Cordyceps sinensis is natural, but Hirsutella sinensis is not natural? Cultivated? – Precise.

Introduction:

The paragraph btw lines: 56 – 86, should be first in the introduction, as it introduce to what exactly the studied object is and why it is important and studied.

The sentence lines 77-79: It has been ……fermentation. – the continuation is needed. “tissues are not just culture medium…” but – what else? In the sentence change “provide” into “provides”

Line 83: fungus/fungi “name” as O. sinensis…..- add “name”

Line 88: Is “sole” proper word? (here and later in the text) – should it be “epi-type” , “type”…or similar.

In the whole “materials and methods” there is lack of crucial information, what analyses were performed right here, by the manuscript authors, in this study. Besides in silico work, if there is any lab work (lines 174-179?) done here, indicate it. Otherwise the reader doesn’t know which material was taken from “references” which was collected during the study.

Line 193 – here and further in the text – I noticed a lot of problems with font size (is not homogenous in many parts of the text), and also with the colour of the font (e.g. green and red in Table 7).

Line 198: All gene…or genes?

Table 2 – these are results of amino-acid properties of what? Precise in the caption.

Results: In my opinion, too many in one paper; there are results for one-two more papers. Authors should consider to choose between the form of presenting some results, because the same information are presented in various forms (table, text, graphic), creating redundant repetitions. Or, consider to put some graphics as supplementary material, this will reduce the volume.

Line 228: Results: Bushley et al 2013 identified….In results section authors should indicate what they done, on what material (providing references….but not starting with the references).  The same: Bushley et al 2013 in line 256…..Zhang et al 2011 in line 269…

Paragraph: 362-377. When you decide to use H. sinensis name be consistent; In the same paragraph is O. sinensis also.

In my opinion, authors should decide which results should be presented: Table 4 of Figure 5 or very detailed description in the text.

Discussion – line 937-939. ..”natural C. sinensis ….independent species“ ….do you mean different strains here? If species – precise, what species.

Figure 12 – should be in results, not in the discussion.

Lines 1087-1097 – this paragraph is “a story of someone else” , and its discussing result obtained by other researchers. In my opinion it is wrong, try to rewrite it to discuss your results, or remove.

Lines 1112-1123 – The name Tolypocladium sinense – is presented here for the first time in manuscript. Where are results for this? Or methods to solve this? The same situation is in line 1151  - 1165, with the Paecilomyces hepiali.

Why Table 7 is in discussion section?

Author Response

Responses to Reviewer 1’s comments:

  1. You commented “in my opinion too extensive material for one publication (only “Results” section has 30 pages!, …there are 95 reference positions). It is really huge work.”

It is true that our study is very large. Our manuscript includes 7 tables and 12 figures. The first submission to BIOLOGY was intended to keep the whole story together for peer review. For the revision, we moved Tables 1–2 and 7 and Figures 1–3 and 12 to the supplementary file to greatly reduce the length of our manuscript.

Per Reviewer 3’s request, we added a new Table 1 in the revision, which lists the technical information of the previously published genome, transcriptome and metatranscriptome sequencing data, thus helping reduce the length of the Methods section.

To respond to your comment on the references, we deleted 17 reference papers, including 11 self-citations of our original research papers.

  1. You questioned “In the title: Cordyceps sinensis is natural, but Hirsutella sinensis is not natural? Cultivated? – Precise.”

We changed the title accordingly to:

Mutations and differential transcription of mating-type and pheromone receptor genes in Hirsutella sinensis and the natural Cordyceps sinensis insect‒fungi complex

“Natural Cordyceps sinensis” refers to the natural insect‒fungi complex, whereas “Hirsutella sinensis” refers to a fungus among the >90 fungal species that have been identified in natural C. sinensis through mycobiota approaches [23,25]. Hirsutella sinensis was postulated to be the sole anamorph (Genotype #1) of Ophiocordyceps sinensis [40]. Molecular studies have identified at least 17 genotypes of O. sinensis [5-6]. We have explained the difficulties and awkward situation of the indiscriminate usage of Latin names in the Introduction section.

  1. You commented “The paragraph btw lines: 56 – 86, should be first in the introduction, as it introduce to what exactly the studied object is and why it is important and studied.”

Following your comment, we have changed the order of the first 2 paragraphs of the Introduction section and moved lines 56–86 (now Lines 48–74 in the revision) to the first paragraph.

  1. You commented “The sentence lines 77-79: It has been ……fermentation. – the continuation is needed. “tissues are not just culture medium…” but – what else? In the sentence change “provide” into “provides”

The whole sentence in our manuscript is as follows:

“It has been suggested that larval tissues are not just culture media that passively provide nutrients for fungal growth, similar to the media used in in vitro culture or fermentation”.

The sentence immediately prior to this sentence is as follows:

“The expression of mating-type genes during the sexual life of natural C. sinensis is much more complex than that in pure fungal cultures and is mutually and/or antagonistically disrupted by the expression of the metagenomes of multiple co-colonized fungi and by larval host innate immunity and acquired immunological responses during O. sinensis fungal infection and proliferation.”

To connect the two sentences, we revised the sentence to respond to your comment as follows:

“It has been suggested that larval tissues are not just culture media that passively provide nutrients for fungal growth, similar to the media used in in vitro culture or fermentation, but also play active defensive roles against fungal infection and growth”.

  1. You commented: Line 83: fungus/fungi “name” as O. sinensis…..- add “name”

Lines 67–71 (now Lines 68–74 in the revision) have been revised as follows:

“In this paper, we refer to the fungus Hirsutella sinensis (Genotype #1) and genetically related Genotypes #2–17 fungi as O. sinensis (note that the 17 genomically independent genotypes share a common genetic ancestor [8], but the taxonomic positions of Genotypes #2 ̶17 have not been determined), and we continue the customary use of the name C. sinensis to refer to the wild or cultivated insect‒fungi complex…”

  1. You commented: Line 88: Is “sole” proper word? (here and later in the text) – should it be “epi-type”, “type” … or similar.

“The sole anamorph” or “the sole teleomorph” in our manuscript refers to the only anamorph or the only teleomorph of O. sinensis and does not mean “epi-type”, “type”, or similar.

Wei et al. [40] proposed “the sole anamorph” and “the sole teleomorph” or “the only anamorph and the only teleomorph” hypotheses for O. sinensis. However, the hypotheses are problematic. For example, the same group of researchers [63] reported the only or sole GC-biased teleomorph (Genotype #1) of O. sinensis in natural C. sinensis but the only or sole AT-biased teleomorph (Genotype #4) of O. sinensis in cultivated C. sinensis. There are at least 2 teleomorphs of O. sinensis because the sequences of Genotype $4, as well as GC-biased Genotypes #2-3 and #7-14 and AT-biased Genotypes #5-6 and #15-17, do not reside in the genome of Genotype #1 H. sinensis and are obviously genomically independent and belong to independent fungi.

Stensrud et al. [8] reported that the GC- and AT-biased genotypes “share a common ancestor” and originated from “accelerated nrDNA evolution” and that the variation among the genotypes “far exceeds what is normally observed in fungi … even at higher taxonomic levels (genera and family)”.

  1. You commented: In the whole “materials and methods” there is lack of crucial information, what analyses were performed right here, by the manuscript authors, in this study. Besides in silico work, if there is any lab work (lines 174-179?) done here, indicate it. Otherwise, the reader doesn’t know which material was taken from “references” which was collected during the study.

All gene, genome, metagenome, transcript, transcriptome and metatranscriptome sequences and other PCR-amplified DNA sequences of the mating-type and pheromone receptor genes of H. sinensis and natural C. sinensis were obtained from public databases and the literature. Our bioinformatic analysis correlates and compares the gene, genome and metagenome sequences with the transcript, transcriptome and metatranscriptome sequences and identifies mutations, differential translations, and alternative splicing of the genes.

Some of the amino acid sequences of the mating-type and pheromone receptor genes used for our bioinformatic analysis are available in public databases. However, we translated many other gene and transcript sequences into protein sequences, especially mutant sequences and transcript sequences with unspliced introns.

In response to your comments, we carefully revised the wording in this section. Per Reviewer 3’s request, we added a new Table 1 in the revision, which includes the technical information for the literature genome, transcriptome and metatranscriptome sequencing according to the corresponding literature. Therefore, the length of the Methods section was greatly shortened.

  1. You commented: Line 193 – here and further in the text – I noticed a lot of problems with font size (is not homogenous in many parts of the text), and also with the colour of the font (e.g. green and red in Table 7).

Thank you for your comment. We standardized the font size in the manuscript across the different sections, and the journal’s production team will ensure the font size in the final version of the paper is homogenous.

  1. You commented: Line 198: All gene…or genes?

Here, we wrote “All gene, genome, transcript, transcriptome and metatranscriptome sequences” in our manuscript. Thus, it is correct to write “All gene … sequences”.

  1. You questioned: Table 2 – these are results of amino-acid properties of what? Precise in the caption.

Table 2 shows the mathematical scaling methods for predicting the secondary structures of proteins on the basis of the physical‒chemical properties of each amino acid component for plotting (at certain window sizes) the α-helices, β-sheets, β-turns, and coils of the proteins according to [56-57]. These plots were generated via the same mathematical scaling method, allowing comparison of the plots for the authentic proteins vs. mutant proteins and prediction of the conformation-related functional alterations of the proteins as the results of point mutations and segment insertion/deletion mutations. Comparisons of the secondary structures of proteins help us and the readers of our paper understand whether mutant proteins maintain their mating and mating communication functions.

Table 2 of the first submission has been moved to the supplementary file as Table S2 to reduce the length of the main text, and its title has been revised to:

Table S2. Amino acid scales for ProtScale analysis (https://web.expasy.org/protscale/) were used to predict the secondary structures (α-helices, β-turns, coils [57] and β-sheets [56]) of the proteins.

ExPASy (https://web.expasy.org/protscale/) defines the ProtScale amino acid scale:

“An amino acid scale is defined by a numerical value assigned to each type of amino acid. The most frequently used scales are the hydrophobicity or hydrophilicity scales and the secondary structure conformational parameter scales, but many other scales exist, which are based on the different chemical and physical properties of the amino acids. This program provides 57 predefined scales entered from the literature.”

We added this definition to the Table legend for Table S2 in the Supplementary file.

  1. You commented: Results: In my opinion, too many in one paper; there are results for one-two more papers. Authors should consider to choose between the form of presenting some results, because the same information are presented in various forms (table, text, graphic), creating redundant repetitions. Or, consider to put some graphics as supplementary material, this will reduce the volume.

We agree that the Results section is long. In the first submission, we were trying to keep the whole story together for peer review. In the revision, we moved Tables 1–2 and 7 and Figures 1–3 and 12 to the supplementary file.

Note that the early versions of Figures 2–3 were published in a Chinese-language paper. However, we found that we cannot clearly compose the whole story without including these 2 figures. After updating the figures by adding more sequence information to display the mutations, we moved the updated Figures 2–3 to the supplementary file as Figures S2–S3.

  1. You commented: Line 228: Results: Bushley et al 2013 identified….In results section authors should indicate what they done, on what material (providing references….but not starting with the references). The same: Bushley et al 2013 in line 256…..Zhang et al 2011 in line 269.

Because of the nature of our bioinformatic study, we must mention the literature in our paper and summarize what has been done in previous studies on the genes, transcripts, and proteins of H. sinensis and natural C. sinensis. The results of our bioinformatic analysis and literature review provide a whole story with the background and logic of our bioinformatic analysis, allowing us to design future laboratory experiments and explore whether mutagenesis is involved in the expression of genes, transcripts, and proteins; whether genes are differentially transcribed and alternatively spliced; and whether mutations, differential transcription, and alternatively spliced protein genes potentially alter the mating and mating communication functions of fungi during the complex sexual life of natural C. sinensis.

We revised the sentences as follows:

Line 228 (now Lines 198-199 in the revision): Three mating-type genes of the MAT1-1 idiomorph have been previously identified in the genome sequence KC437356 of the H. sinensis strain 1229 by Bushley et al. [37]…

Lines 256–259 (now Lines 223–227 in the revision): The presence of multicellular heterokaryotic structures of C. sinensis hyphae and ascospores was detected via fluorescence microscopy techniques in natural C. sinensis by Bushley et al. [37]. These authors also reported that the MAT1-1 and MAT1-2 idiomorphs are not closely linked because the MAT1-1-1 and MAT1-2-1 genes are located more than 4 kb apart in the H. sinensis genome.

Lines 282–285 (now Lines 88–90 in the revision): … Zhang and Zhang [39] reported differential occurrence of the mating-type genes of the MAT1-1 and/or MAT1-2 idiomorphs in various C. sinensis isolates and hypothesized facultative hybridization that H. sinensis undergoes facultative hybridization.

Lines 470–471 (now Section 3.4.2, Lines 376–377 in the revision): Zhang et al. [64] conducted a metatranscriptome study in natural C. sinensis at the early maturing stage and reported differential transcription of mating-type genes.

Lines 480–482 (now Section 3.4.2, Lines 386–388 in the revision): Zhong et al. [65] conducted another metatranscriptome study of the early-stage stroma and caterpillar body of natural C. sinensis collected from Yushu, Qinghai Province, China, and observed differential transcription of mating-type genes.

Lines 490–494 (now Section 3.4.2, Lines 397–406 in the revision): Li et al. [66] conducted a metatranscriptome study in cultivated C. sinensis at all developmental phases and observed the differential transcription of the mating-type genes of the MAT1-1 and MAT1-2 idiomorphs at very low levels (with read counts of 0‒40 and fragments per kilobase of exon model per million mapped fragments [FPKM] values of 0‒8.4) (note: an FPKM value less than 10 indicates very low levels of transcription).

Lines 499–502 (now Section 3.4.2, Lines 406–410 in the revision): Zhao et al. [67] conducted a metatranscriptome study in cultivated C. sinensis and reported nearly no transcription of the mating-type genes of the MAT1-1 and MAT1-2 idiomorphs (transcripts per million reads [TPM] of 0–2.27).

  1. You commented: Paragraph: 362-377. When you decide to use H. sinensis name be consistent; In the same paragraph is O. sinensis also.

We have carefully selected and used the names H. sinensis and O. sinensis in our paper, although some mycological scholars believe that these 2 names are unconditionally interchangeable. In recent years, molecular studies have demonstrated that H. sinensis is GC-biased Genotype #1 of O. sinensis, which is the only genotype that has been taxonomically characterized. Genotypes #2–17 are all under the name of O. sinensis in GenBank and numerous papers but, unfortunately, have not been purified and taxonomically determined. The sequences of Genotypes #2–17 have not been found in the genome assemblies of 5 H. sinensis strains and are clearly genomically independent, belonging to independent fungal species “even at higher taxonomic levels (genera and family)”, as stated by Stensrud et al. [8]. Thus, O. sinensis in our study refers to more than one genotype or all 17 genotypic fungi of O. sinensis.

We wrote (Lines 366–369) (now Section 3.2, Lines 292–304 in the revision) that “Instead, these findings suggest that O. sinensis requires sexual partners from H. sinensis, monoecious or dioecious, for physiological heterothallic reproduction or from heterospecific fungal species to ensure hybridization if the species are able to overcome interspecific reproductive isolation.

The available evidence regarding mating-type gene transcription might explain why previous efforts to cultivate pure H. sinensis in research-oriented academic settings to induce the production of fruiting bodies and ascospores have consistently failed [36,49,61-62]. The evidence may also explain why a successful inoculation–cultivation project in a product-oriented industrial setting presented a species contradiction between the GC-biased Genotype #1 H. sinensis strains used as anamorphic inoculants and the teleomorphic AT-biased Genotype #4 of O. sinensis in the fruiting body of cultivated C. sinensis, as reported by Wei et al. [63].

We intentionally stated that “O. sinensis requires sexual partners” here and in other places in our paper because all 17 genomically independent genotypes, either GC- or AT-biased, are under the name “O. sinensis” without taxonomic determinations of Genotypes #2–17 and because our study reported that self-sterile H. sinensis (GC-biased Genotype #1) is not able to accomplish sexual reproduction alone. Other genotypes, either GC- or AT-biased, of O. sinensis may participate in the sexual reproduction of O. sinensis. Coincidentally, several studies [7,16,41-42,63] reported that one of AT-biased genotypes occurred in natural C. sinensis without the co-occurrence of GC-biased Genotype #1 H. sinensis. Thus, the use of “O. sinensis” in our paper indicates that sexual reproduction of O. sinensis requires not only the GC-biased Genotype #1 H. sinensis but also one or more of the genomically independent Genotypes #2–17 of O. sinensis fungi.

Moreover, Barseghyan et al. [24] reported that Tolypocladium sinensis and H. sinensis are dual anamorphs of O. sinensis, suggesting that the 2 anamorphs may participate in the sexual reproduction of O. sinensis. Engh [41] reported that Cordyceps and Tolypocladium form a natural complex in natural C. sinensis, where “Cordyceps” was identified as AT-biased Genotype #4 of O. sinensis [42]. Regardless of whether the hypotheses are correct, we have no position in the academic debate. However, carefully selecting the name “O. sinensis” is the only suitable choice for describing the unknown sexual reproduction of O. sinensis in the sexual life of natural C. sinensis insect‒fungi complexes.

  1. You commented: “In my opinion, authors should decide which results should be presented: Table 4 of Figure 5 or very detailed description in the text”.

Table 4 (now Table 3 in the revision) lists the percentages of hydrophobic, acidic, basic, and neutral amino acid components of the MAT1-1-1 and MAT1-2-1 proteins of H. sinensis and natural C. sinensis, presenting the overall profiles of the hydrophobic properties of the proteins. Figure 5 (now Figure 2 in the revision) presents the regional hydrophobicity‒hydrophilicity properties of small segments of proteins sequentially at a window of 21 amino acids. Both Table 4 and Figure 5 (now Table 3 and Figure 2 in the revision) allow comparison of the overall and sequentially regional hydrophobicity‒hydrophilicity properties over the entire lengths of the authentic and mutant proteins. Thus, Table 4 and Figure 5 predict the secondary structures of the proteins from different perspectives.

Similarly, Table 6 and Figure 11 (now Table 5 and Figure 8 in the revision) present the overall and sequential regional hydrophobicity‒hydrophilicity properties of the proteins. Thus, both Table 6 and Figure 11 are important.

The percentage differences in amino acid components between Tables 4 and 6, particularly the differences in hydrophobic amino acids, indicate that the pheromone receptor proteins are membrane-bound proteins receiving mating signals from pheromone proteins of different fungal cells, whereas the MAT1-1-1 and MAT1-2-1 proteins are likely secretory proteins. Information on protein secondary structures will help scientists understand the nature of proteins and design future studies in protein biochemistry, cell biology, reproductive physiology, and other scientific disciplines.

  1. You questioned: Discussion – line 937-939. “natural C. sinensis …. independent species“ ….do you mean different strains here? If species – precise, what species.

No. The term “independent species” here does not mean “different strains”. To respond to your comment, we changed “independent species” to “independent fungal species”.

Stensrud et al. [8] analyzed for the first time the 71 ITS sequences of O. sinensis and divided the O. sinensis sequences into 3 groups, Groups A–C, along with fungal sequences other than O. sinensis. These authors believe that Groups A–C of O. sinensis share a common genetic ancestor, but the genetic differences among the O. sinensis groups “far exceed what is normally observed in fungi (and other organisms), even at higher taxonomic levels (genera and family).” Group A is GC-biased Genotype #1 (H. sinensis), and Groups B–C are AT-biased Genotypes #4–5 of O. sinensis.

Since 2007, many studies in the field of C. sinensis research have identified additional genotypes of O. sinensis from natural C. sinensis samples collected from different production areas in China, India and Nepal. Among the 17 genotypes, including GC-biased Genotypes #1–3 and #7–14 and AT-biased Genotypes #4–6 and #15–17, only Genotype #1 has been taxonomically characterized as H. sinensis. Unfortunately, no reports to date are available on pure cultures of Genotypes #2–17 of O. sinensis, preventing taxonomical determination, multigene and multilocus analyses, and genome and transcriptome sequencing of Genotypes #2–17. However, the sequences of Genotypes #2–17 do not reside in the 5 genome assemblies of H. sinensis strains, which very likely belong to independent fungal species “even at higher taxonomic levels (genera and family)”, as stated by Stensrud et al. [8]. Purification and taxonomic identification of Genotypes #2–17 fungi will rely on mycologists and mycology taxonomists and are far beyond our expertise. Thus, prior to determining the taxonomic positions of Genotypes #2–17, we have no other option but to habitually follow the annotations in GenBank and the literature and label the genomically independent Genotypes #2–17 as O. sinensis fungi, which means that O. sinensis is not a single fungal species but instead multiple independent fungal species. We hope that mycological taxonomists will be able to end this indiscriminate but habitual practice by replacing the name with unique and exclusive names for Genotypes #2–17 fungi.

  1. You commented: Figure 12 – should be in results, not in the discussion.

An early version of Figure 12 was published previously in a Chinese-language journal (Journal of Fungal Research 2012, 10(2): 100–112. http://en.cnki.com.cn/Article_en/CJFDTotal-YJJW201202007.htm). This paper reports the co-occurrence of GC-biased Genotypes #1 and #2, and their abundances were altered significantly and dynamically in an asynchronous, disproportional manner in the stromata of natural C. sinensis during maturation, indicating the genomic independence of the 2 GC-biased genotypes. Further studies reported that the sequences of Genotype #2 do not reside in the genome of Genotype #1 H. sinensis but belong to an independent O. sinensis fungus. Unfortunately, this Chinese-language paper may not be accessible or understandable to English-speaking readers in the international scientific community.

Because our current bioinformatic study (Bioogy-2963648) revealed that H. sinensis (GC-biased Genotype #1 of O. sinensis) is self-sterilizing and that the sexual reproduction of O. sinensis requires mating partners, the co-occurrence of GC-biased Genotypes #1–2 in the stromata of natural C. sinensis during maturation may provide important information related to the mating process under heterothallism (if Genotypes #1–2 belong to the same fungal species) or hybridization (if Genotypes #1–2 are different fungal species). With permission from the previous publisher, we reproduced the figure with modifications and added it to the Discussion section to inform English-speaking readers in the international scientific community about the co-occurrence of the 2 GC-biased genotypes in the stromata of natural C. sinensis during maturation. Because our manuscript is too long, we have deleted this figure but retained the description in Section 4.2.2 of the Discussion.

  1. You commented: Lines 1087-1097 – this paragraph is “a story of someone else”, and its discussing result obtained by other researchers. In my opinion it is wrong, try to rewrite it to discuss your results, or remove:

Your comment referred to the following paragraph (Lines 1087–1097) (now Section 4.2.6, Lines 952–964 in the revision):

Hu et al. [36] reported the use of a mixture of two pure H. sinensis strains, Co18 and QH195-240, to inoculate 40 larvae of Hepialidae sp. Fungal inoculation induced death and mummification of the larvae but failed to induce the development of fruiting bodies and ascospores, indicating biological separation of the larval death/mummification process and the fungal fruiting body development process. The authors cited two other studies [61-62] and reported that inoculation of ghost moth larvae of the Hepialidae family with pure H. sinensis consistently failed to produce fruiting bodies and ascospores. Zhang et al. [49] (coauthors of [36]) summarized 40 years of experience in artificial cultivation of C. sinensis and concluded that “it is very difficult in our laboratory to induce development of the C. sinensis fruiting bodies, either on culture medium or on insects.”

Lines 1025–1034 (now Section 4.2, Lines 885–894 in the revision) discuss previous findings that H. sinensis alone is not able to accomplish sexual reproduction. Several citations have been provided to support this general statement, including (6) (now Section 4.2.6, Lines 952–964  in the revision) in the Discussion section, as you noted. The last sentence of this paragraph discusses Zhang et al.’s paper [49], which summarized 40 years of scientific experiments, including the findings of Holliday & Cleaver [61], Stone [62], and Hu et al. [36]. In this work, we restate the summary of Zhang et al. [49], which supports our self-sterility hypothesis for H. sinensis. Additionally, our mutation and differential transcript findings for the mating-type and pheromone receptor genes in H. sinensis provide explanatory evidence at the genetic, transcriptional, and protein levels for the failure of sexual reproduction experiments in H. sinensis.

To respond to your comment, we revised the paragraph by adding one sentence at the end of the paragraph as follows:

4.2.6 Hu et al. [36] reported the use of a mixture of two pure H. sinensis strains, Co18 and QH195-240, to inoculate 40 larvae of Hepialidae sp. Fungal inoculation induced death and mummification of the larvae but failed to induce the development of fruiting bodies and ascospores, indicating biological separation of the larval death/mummification process and the fungal fruiting body development process. The authors cited two other studies [61-62] and reported that inoculation of ghost moth larvae of the Hepialidae family with pure H. sinensis consistently failed to produce fruiting bodies and ascospores. Zhang et al. [49] (coauthors of [36]) summarized 40 years of experience in artificial cultivation of C. sinensis and concluded that “it is very difficult in our laboratory to induce development of the C. sinensis fruiting bodies, either on culture medium or on insects.” Our findings presented in this paper provide evidence at the genetic, transcriptional, and protein levels to explain the self-sterility of H. sinensis and the failure of the inoculation experiments using pure cultures of H. sinensis as the sole inoculant.

  1. You commented: Lines 1112-1123 – The name Tolypocladium sinense – is presented here for the first time in manuscript. Where are results for this? Or methods to solve this? The same situation is in line 1151-1165, with the Paecilomyces hepiali.

Our study does not include analyses of the gene and transcript sequences of Tolypocladium sinense (Lines 1112–1123) (now Section 4.2.8, Lines 981–995 in the revision) because of the limited information on T. sinensis in public databases. However, T. sinense or T. sinensis has been a mystery of natural C. sinensis since Prof. Li Chao-Lan identified, purified, and taxonomically determined T. sinense for the first time in 1988 [17]. Engh [41] reported that T. sinensis and C. sinensis form a natural fungal complex in which the “C. sinensis” was subsequently identified as AT-biased Genotype #4 [42]. Barseghyan et al. [24] reported that T. sinensis and H. sinensis are dual anamorphs of O. sinensis. Quandt et al. [2014; IMA Fungus 5(1): 121–134] revised the family Ophiocordycipitaceae and added the genus Tolypocladium as a new member of Ophiocordycipitaceae.

As a follow-up to the reporting of the self-sterility of H. sinensis in our bioinformatic paper, the scientific community may need to search for the mating partner(s) of O. sinensis. The close association between T. sinensis and H. sinensis may make it a candidate for future sexual reproduction physiology studies. Thus, we added the following sentence at the end of this paragraph:

4.2.8 Tolypocladium sinense in natural C. sinensis was first identified and reported by Li [17]. It was subsequently isolated from natural C. sinensis and characterized morphologically and genetically [20-21]. Engh [41] reported the molecular identification of the Cordyceps‒Tolypocladium complex in natural C. sinensis. The “Cordyceps” sequence AJ786590 obtained by Engh [41] was published and uploaded to GenBank by Stensrud et al. [42] and phylogenetically clustered into AT-biased Group B (Genotype #4) of O. sinensis, along with other C. sinensis sequences, by Stensrud et al. [8]. Barseghyan et al. [24] performed a macro/micromycology study and concluded that H. sinensis, which is presumed to be psychrophilic, and T. sinensis, which is presumed to be mesophilic, are dual anamorphs of O. sinensis. Notably, the O. sinensis fungus, which has H. sinensis-like morphology and growth characteristics, was not genotyped molecularly in that study. According to the self-sterility hypothesis for H. sinensis presented in this bioinformatic paper, the close association of T. sinense with H. sinensis may help mycological physiologists plan future studies to explore the possibility of O. sinensis hybridization reproduction.

Similarly, discussion section (7) (now Section 4.2.7 in the revision, as well as Sections 4.2.1, 4.2.4, and 4.2.5) discusses the possibility of hybridization of GC-biased H. sinensis with one of the AT-biased genotypes of O. sinensis if they belong to different fungal species “even at higher taxonomic levels (genera and family)”. Sections (9) and (10) (now Sections 4.2.9 and 4.2.10, respectively) discuss the possibility of hybridization or parasexuality of GC-biased H. sinensis with an AB067719-type fungus or Paecilomyces hepiali under the self-sterility hypothesis for H. sinensis. We have added a short sentence at the end of Sections 4.2.9 and 4.2.10.

  1. You questioned: Why Table 7 is in discussion section?

Table 7 provides information to support Section (9) (now Section 4.2.9) of the Discussion section. This table was published in our prior publication, PLoS ONE. 2023;18: e0270776. DOI: 10.1371/journal.pone.0270776, and demonstrated that two GC-biased genotypes, #13 and #14 of O. sinensis, feature precise reciprocal substitutions of large DNA segments and genetic material recombination between two parental fungi, Genotype #1 H. sinensis (Group A by Stensrud et al. [8]) and an AB067719-type Group E fungus. These genetic features suggest sexual reproduction hybridization or parasexuality, representing a perfect example for discussing the sexual reproduction model(s) of natural C. sinensis.

Because of the length of our paper, we moved this table to the supplementary file as Table S3.

Reviewer 2 Report

Comments and Suggestions for Authors

The article presents too much data, does not comply with the MDPI standards and also presented a high percentage of plagiarism, greater than 40%, so the article must be rejected.

Comments on the Quality of English Language

Extensive editing of English language required

Author Response

Responses to Reviewer 2’s comments:

  1. You commented: The article presents too much data, does not comply with the MDPI standards

We have moved Tables 1-2 and 7 and Figures 1-3 and 12 to the supplemental files to greatly reduce the length of our paper.

  1. (The article) presented a high percentage of plagiarism, greater than 40%.

You commented that our manuscript “presented a high percentage of plagiarism, greater than 40%”. However, you did not provide evidence of plagiarism and the reference papers that have been formally published by other research groups and that you used for plagiarism checks.

The editorial office of BIOLOGY provided us with results of duplicate checks, which revealed a high percentage of similarity (repeats or duplicates). We then were required to thoroughly analyze the duplicate-check results and the references that were used in the duplicate-check. However, when the preprints and conference presentation-paper of our own works from the list of duplicate-check references are omitted, the total similarity is only approximately 14% or less. This very low similarity still includes numerous “duplicates” of authors’ names, university names, province names, country names, “*” signs showing the corresponding author, email addresses of the corresponding author; the word “the” and words “It has been”, “and the”, “for the” and “of the”; numerous citations of literature papers, including the "et al.", “al. 2014” and publishing years; as well as public information such as GenBank accession numbers and fungal strain numbers, sample collection locations, and even hyphens in sequence alignment figures, small segments of protein sequences, and many more, as shown below, with colorfully highlighted “duplicates”, which resulted from the duplicate check.

The so-called "duplicates" shown in the colorfully highlighted, duplicate-checked manuscript are too numerous to enumerate, containing a total of 1308 lines in our paper.

It is common sense and a consensus in the scientific community worldwide that preprints and conference presentation-paper do not count as formal publications and do not hinder their official publication in journals. This common sense has been widely accepted and practiced by the international scientific community.

Plagiarism is defined as taking someone else’s work or ideas and claiming them as your own. Because the same group of authors (Li X-Z, Li Y-L, and Zhu J-S) published preprints and conference presentation-paper that were used as references for duplicate checks and accounted for a total of 34-36% similarity, our current manuscript (Biology-2963648) does not contain incidences of plagiarism. Thus, your comment that “a high percentage of plagiarism, greater than 40%” was apparently groundless, and our manuscript was mistakenly accused.

A preliminary study was published in a Chinese language journal《中国中药杂志China Journal of Chinese Materia Medica 2023, 48(10): 2829–2840), which discussed the differential occurrence and transcription of mating-type genes in natural C. sinensis and in pure cultures of H. sinensis. However, the vast majority of English-speaking scholars in the international scientific community are unable to read this Chinese-language paper. Some Chinese-speaking scholars continue to state that H. sinensis is self-fertilized and uses homothallism or pseudohomothallism for sexual reproduction. Thus, it is necessary for our current manuscript (Biology-2963648) to include the preliminary data presented in the Chinese-language paper along with more updated data, as well as extensive bioinformatic analysis.

This preliminary Chinese language paper contains early versions of Figures 1-2 showing the alignments of the MAT1-1-1 and MAT1-2-1 gene sequences. These 2 figures have been largely updated and modified by adding translated protein sequences and more relevant sequences to demonstrate mutations at the genetic, transcriptional, and protein levels for the current manuscript (Biology-2963648) and were included as Figures 2-3 in the first submission. Because the current manuscript contains too many tables and figures, we moved the largely updated figures to the supplementary files as Tables S2-S3.

  1. Regarding your comment on "the Quality of English Language. Extensive editing of English language required":

Our manuscript has undergone extensive English language editing during the past 3 years by an academic language editing firm (American Journal of Experts or AJE) by both an AI editing program and at least 2 senior human editors. Please see the AJE editing certificate below. Our manuscript revision has also been revised by the same group of editors:

Reviewer 3 Report

Comments and Suggestions for Authors

The study is about the sexual reproduction mechanism of Hirsutella sinensis focusing on the role of mating-type loci. The researchers compared MAT1-1 and MAT1-2 in 237 strains and 5 different genotypes of H. sinensis. The differential occurrence and transcription of these genes suggest self-sterility of H. sinensis and possible crossbreeding between different fungal types.

Some figures span multiple pages. It might be worthwhile to either restructure them or move them to a supplementary file. 

Methods and other stats about the assemblies done by the authors should be reported the main text. Methods and stats about previously published data can be put into a table. 

Line 192-196: Missing data makes it impossible to replicate these steps.

Additional Comments:

The study primarily investigates variation in mating-type genes and pheromone receptor genes in existing data to draw conclusions about sexual reproduction of H. sinensis and O. sinensis. The analysis of MAT locus via comparison of existing published genomes/sequence data is relevant for the field. The study presents a hypothesis, that H. sinensis might be crossbreeding, based on the evidence of differential occurrence and transcription of MAT genes and lack of evidence supporting self fertilization.

This study adds to the field by challenging the existing hypotheses about self-fertilization and pseudohomothallism in H. sinensis.

The study did not involve direct experimental activities by the authors. Instead, it relied on the analysis of existing data to present new hypothesis about the genetics and reproduction strategies. Experimental work involving controlled mating experiments, would be beneficial to directly test these new hypotheses​.

The conclusions drawn by the authors about mechanisms of sexual reproduction in H. sinensis are based on the observed patterns in the sequence data and the transcriptional behavior of MAT genes. The conclusions are based on correlations observed in existing datasets rather than controlled experiments.

The study uses existing data to propose hypotheses about reproductive strategies, but it does not provide experimental verification of these hypotheses. Authors should note this in their conclusion section.

The number of references seems a bit long. Tens of papers cited for one point is not useful.

Table 1: Lists different strains and occurrence of MAT genes. This can be moved to supplementary.

Table 2: Not sure if this is useful. It needs more explanation about how this information was used. Figures 2, 3, 4, 6, 7, 8, 9, 10 show alignments. Authors need to pick and focus on a few alignments, and move the rest to supplementary.

Figures 5 and 11: Span multiple pages. These need to be converted to composite figures on a single page each.

Authors need to focus on their methods and the results they provide rather than detailing methods from other studies. Their own results are getting lost in the details of other studies. The discussion about the results of existing literature make it more of a review article.

Author Response

Responses to Reviewer 3’s comments:

  1. You commented: “Some figures span multiple pages. It might be worthwhile to either restructure them or move them to a supplementary file.”

Figure 5 (now Figure 2 in the revision) examines the protein sequences of 3 samples, whereas Figure 11 (now Figure 8 in the revision) examines 4 samples. The samples were analyzed for α-helices, β-sheets, β-turns, and coils to predict the secondary structures of the proteins. Comparisons of the α-helix, β-sheet, β-turn, and coil plots were based on regional alterations in the hydrophobicity‒hydrophilicity properties of amino acid components sequentially at a window of 21 amino acids in the proteins. The comparisons allowed us to predict possible conformational alterations of authentic and mutant proteins and indicate possible alterations in the mating and mating communication functions of the proteins. The length of the manuscript can be reduced if the figure layout is designed as double columns to accommodate Figures 5 and 11 (now Figures 2 and 8 in the revision).

We reorganized the plots for multiple samples with waveform alignment (vertically) among the sample plots and moved the original plots to the supplementary files, as shown in Figures S4-S5. Therefore, Figures 5 and 11 (now Figures 2 and 8 in the revision) each have 4 panels for the α-helices, β-sheets, β-turns, and coils; each panel contains 3 or 4 ExPASy ProtScale plots for different samples.

Exploration of the conformational changes in the authentic and mutant proteins is crucial to the main subject of this paper, which indicates possible functional alterations in the sexual reproduction processes of O. sinensis during the sexual life of natural C. sinensis. Thus, we need to leave Figures 5 and 11 (now Figures 2 and 8 in the revision) in the main body of our paper.

  1. You commented “Methods and other stats about the assemblies done by the authors should be reported the main text. Methods and stats about previously published data can be put into a table.”

Thank you for your suggestion. We added a new table (Figure 1 in the revision) listing the sequencing and assembling technologies (software) for the genome, transcriptome, and metatranscriptome assembly sequences that have been previously published and uploaded to GenBank and other repository databases. Therefore, this change has greatly shortened the Methods section of our manuscript.

  1. You commented “Line 192-196: Missing data makes it impossible to replicate these steps.”

Lines 192–196 (now Section 2.2, Lines 170–173 in the revision): “The metatranscriptome assembly sequences were uploaded to the data repository, www.plantkingdomgdb.com/Ophiocordyceps_sinensis/data/cds/Ophiocordyceps_sinensis_CDS.fas, which is unfortunately currently inaccessible, but a previously downloaded cDNA file was used for the bioinformatic analysis.”

This website did not become inaccessible until 2021 or later.

We may provide the prior downloaded PDF file as a private reference for a special review upon a written inquiry from the editorial office and reviewers. Because of copyright protection, however, we cannot disclose the PDF cDNA document and other documents to the public.

  1. You commented: “The study primarily investigates variation in mating-type genes and pheromone receptor genes in existing data to draw conclusions about sexual reproduction of H. sinensis and O. sinensis. The analysis of MAT locus via comparison of existing published genomes/sequence data is relevant for the field. The study presents a hypothesis, that H. sinensis might be crossbreeding, based on the evidence of differential occurrence and transcription of MAT genes and lack of evidence supporting self-fertilization.

This study adds to the field by challenging the existing hypotheses about self-fertilization and pseudohomothallism in H. sinensis.

The study did not involve direct experimental activities by the authors. Instead, it relied on the analysis of existing data to present new hypothesis about the genetics and reproduction strategies. Experimental work involving controlled mating experiments, would be beneficial to directly test these new hypotheses.

The conclusions drawn by the authors about mechanisms of sexual reproduction in H. sinensis are based on the observed patterns in the sequence data and the transcriptional behavior of MAT genes. The conclusions are based on correlations observed in existing datasets rather than controlled experiments.

The study uses existing data to propose hypotheses about reproductive strategies, but it does not provide experimental verification of these hypotheses. Authors should note this in their conclusion section.”

We stated the following in the last sentence of the Introduction section:

Numerous genetic, genomic and transcriptomic sequences of H. sinensis strains and natural C. sinensis specimens are available in the GenBank database, which enables further bioinformatic examination of the hypotheses regarding the sexual reproduction strategy of O. sinensis at the genome, transcriptome, and protein levels.

We also stated this in the first sentence of the Conclusion section as follows:

Bioinformatic analysis of genomic, transcriptomic and protein sequences available in public databases revealed the differential transcription of the mating-type genes of the MAT1-1 and MAT1-2 idiomorphs and pheromone receptor genes of H. sinensis.

We believe that in silico analysis by cross-referencing and integrating all available public data regarding the genomic, transcriptomic, metatranscriptomic, and protein sequences of H. sinensis and natural C. sinensis is crucial for understanding the full view of the sexual reproduction model(s) of O. sinensis and for designing future laboratory experiments. Aligning these sequences helps scientists understand mutations in genes, transcripts, and proteins and the differential transcription and alternative splicing of genes to design future laboratory experiments. Additionally, changes in the overall hydrophobicity of proteins and sequential regional hydrophobicity alterations in protein sequences will help predict possible changes in the secondary structures of proteins caused by mutagenesis. All these bioinformatic analytical results constitute the foundation of future study design beyond the genome, transcriptome and metatranscriptome sequences. In other words, scientists may not be able to correctly formulate future study directions and, in contrast, may go in the wrong direction in their studies if they do not fully understand the integrated bioinformatic analytical results.

For example, the literature [36-37] has suggested self-fertilization for H. sinensis under homothallic and pseudohomothallic sexual reproduction. These hypotheses have been widely adopted internationally in academic fields. Under the incorrect hypothesis, Hu et al. [36] failed to induce the development of C. sinensis fruiting bodies in their artificial inoculation experience using H. sinensis as the sole inoculant on 40 larvae of Hepialus spp. However, we conducted this bioinformation study using the same set of data with cross-referencing and integrating all the related data but drew completely different conclusions, namely, self-sterility for H. sinensis. Hu et al. [36] and Bushley et al. [37] might have misinterpreted some important information from their own datasets and the public literature data. It is no surprise that Zhang et al. [49] (coauthors of [36]) summarized 40 years of experience in artificial cultivation of C. sinensis, presumptively including the use of H. sinensis as the sole inoculant, and concluded that “it is very difficult in our laboratory to induce development of the C. sinensis fruiting bodies, either on culture medium or on insects”. However, scientists may avoid such experimental failure after reading and fully understanding the results of our bioinformatic analysis.

  1. You commented: “The number of references seems a bit long. Tens of papers cited for one point is not useful.”

We have deleted 16 references from the list, including 11 self-citation references, largely reducing the list to only 78 references in the revision, which is not too many for a bioinformatics project.

  1. You commented “Table 1: Lists different strains and occurrence of MAT genes. This can be moved to supplementary.”

Yes. We followed your suggestion and moved Table 1 to the supplementary file as Table S1.

  1. You commented “Table 2: Not sure if this is useful. It needs more explanation about how this information was used. Figures 2, 3, 4, 6, 7, 8, 9, 10 show alignments. Authors need to pick and focus on a few alignments, and move the rest to supplementary.”

Table 2 in the first submission provides the key reference information for mathematically computing the segment hydrophobicity in protein sequences sequentially at a window of 21 amino acid residues for predicting the secondary structures along the protein sequences. The mathematical method was originally developed by Chou & Fasman [56] and Deleage & Roux [57]. The key references presented in Table 2 for each of the amino acid components provide the mathematical basis of the ExPASy ProtScale algorithm (https://web.expasy.org/protscale/) for computing the sequential regional properties and scales of the proteins and exploring the secondary structures (conformations) of the proteins for planning future laboratory experiments.

Gasteiger et al. [2005; Protein Identification and Analysis Tools on the Expasy Server. Chapter 52 (In) John M. Walker (ed): The Proteomics Protocols Handbook, Humana Press (2005). pp. 571–607] reported that “For protein analysis, information in protein databases can be used to predict certain properties about a protein, which can be useful for its empirical investigation”.

Because Table 2 provides the technical information and mathematical analysis tools related to sequential segmental hydrophobicity for predicting the secondary structures of proteins, we retained the table in the supplementary file to save space in the main text.

You also commented on “Figures 2, 3, 4, 6, 7, 8, 9, 10”.

Figure 2: Alignment of the MAT1-2-1 gene sequences with the transcriptome sequences

Figure 3: Alignment of the MAT1-1-1 gene sequences with the transcriptome sequences

Figure 4: Alignment of the MAT1-1-1 protein sequences

Figure 6: Alignment of the MAT1-2-1 protein sequences

Figure 7. Alignment of the genome and transcriptome sequences of the a-pheromone receptor

Figure 8. Alignment of the protein sequences of the a-pheromone receptor

Figure 9. Alignment of the genome and transcriptome sequences of the α-pheromone receptor

Figure 10. Alignment of the protein sequences of the α-pheromone receptor

All these figures are important to the main subject of our bioinformatics paper. However, because the early versions of Figures 2–3 were published in a preliminary report in a Chinese-language journal (Li et al. 2023a), we moved Figures 2–3, which contain updated information and data, to the supplementary file as Figures S2–S3. The remaining figures (Figures 4, 6–10) are included in the main text, now Figures 1, 3–7 in the revision.

  1. You commented “Figures 5 and 11: Span multiple pages. These need to be converted to composite figures on a single page each.”

Figures 5 and 11: ExPASy ProtScale plots for the α-helices, β-sheets, β-turns, and coils of the MAT1-1-1 proteins and α-pheromone receptor proteins, respectively.

To help our readers better understand the scientific meaning of ExPASy ProtScale plots in comparisons of the authentic and mutant proteins, we reorganized the plots with waveform alignment (vertically) among the sample plots and moved the original plots to the supplementary file as Figures S4–S5. Therefore, Figures 5 and 11 (now Figures 2 and 8 in the revision) each have 4 panels for the α-helices, β-sheets, β-turns, and coils; each panel contains 3 or 4 ProtScale plots.

The use of the ExPASy ProtScale plotting technique for the comparison of the secondary structures of authentic and mutant proteins in the field of mycology is relatively new. We have tried to composition plots into fewer panels as an option in handling multiple plots, but this practice was not successful because of the topological complexity of the plot waveforms.

  1. You commented “Authors need to focus on their methods and the results they provide rather than detailing methods from other studies. Their own results are getting lost in the details of other studies. The discussion about the results of existing literature make it more of a review article.”

The Discussion section of our bioinformatic study contains analyses of the literature. However, all the cited literature and relevant analyses are related to the main subject of our bioinformatic paper, namely, fungal mating processes, the sexual reproduction of O. sinensis, and the self-sterility of H. sinensis.

For example, the following objective literature findings are crucial to accomplish the sexual reproduction of O. sinensis during the sexual life of natural C. sinensis on the basis of the self-sterility of H. sinensis and the requirements of mating partners:

Lines 1035–1043 and 1064–1086 (now Sections 4.2.1, 4.2.4, and 4.2.5, Lines 895–908 and 929–951 in the revision): Differential co-occurrence of GC-biased H. sinensis and AT-biased O. sinensis in the stromata, stromal fertile portion, ascocarps, and ascospores of natural C. sinensis.

Lines 1044–1052 (now Section 4.2.2, Lines 909–919 in the revision): GC-biased Genotypes #1 and #2 of O. sinensis co-occur in the stromata of natural C. sinensis, and their dynamic alteration in abundance occurs in an asynchronous and disproportional manner during C. sinensis maturation.

Lines 1057–1063 (now Section 4.2.3, Lines 920–928 in the revision): GC-biased Genotypes #1 and #7 of O. sinensis co-occurrence in different compartments of a single natural C. sinensis specimen.

Lines 1087–1097 (now Section 4.2.6, Lines 952–964 in the revision): Failure of artificial cultivation using GC-biased Genotype #1 H. sinensis as the inoculant.

Lines 1098–1111 (now Section 4.2.7, Lines 965–980 in the revision): Identification of two teleomorphs of O. sinensis: GC-biased Genotype #1 in natural C. sinensis and AT-biased Genotype #4 in cultivated C. sinensis in an industrial artificial inoculation study.

Lines 1112–1123 (now Section 4.2.8, Lines 981–995 in the revision): Close relationships in the natural complex, possibly even dual anamorphs, of Tolypocladium sinensis and AT-biased Genotype #4 of O. sinensis in natural C. sinensis.

Lines 1124–1145 (now Section 4.2.9, Lines 996–1016 in the revision): Identification of GC-biased Genotypes #13–14 of O. sinensis in the ascospores of natural C. sinensis, which are the offspring of two parental fungi, H. sinensis and an AB067719-type fungus. Genotypes #13–14 of O. sinensis feature precise reciprocal substitutions of large DNA segments of the parental fungi and genetic material recombination. Genetic features indicate the outcomes of fungal hybridization or parasexuality.

Lines 1151–1165 (now Section 4.2.10, Lines 1017–1031 in the revision): P. hepiali is closely associated with GC- and AT-biased genotypes of O. sinensis in all compartments of natural C. sinensis, including the ascocarps and ascospores of natural C. sinensis. The isolation and purification of P. hepiali from the wide-type O. sinensis fungal complex is difficult even for the most experienced senior mycologists, indicating a close physiological relationship between P. hepiali and genotypes of O. sinensis. Studies [5,14-15] have demonstrated that P. hepiali participates in the ejection process of ascospores (sexual cells) from ascocarps (sexual organs) of natural C. sinensis.

Thus, the literature and the relevant analysis of the objective findings listed in the Discussion section are not a randomized literature review but are closely related to the main subject of our bioinformatic paper: the mating processes and sexual reproduction of O. sinensis in natural C. sinensis. Mycology scientists might not have integrated these pieces of objective data and information into the mating processes and sexual reproduction of O. sinensis, but our paper focuses on these objective findings for mycologists and reproduction physiologists by integrating all the pieces of information, many of which have been nearly forgotten and are unfamiliar to the majority of scientists in the international scientific community, under the self-sterility hypothesis for H. sinensis during the sexual life of natural C. sinensis.
